# CK-666 and CK-869 differentially inhibit Arp2/3 iso-complexes

LuYan Cao [ID] [1✉], Shaina Huang[1], Angika Basant[1], Miroslav Mladenov [ID] [1] & Michael Way [ID] [1,2✉]

## Abstract

**The inhibitors, CK-666 and CK-869, are widely used to probe the function of Arp2/3 complex mediated actin nucleation in vitro and in cells. However, in mammals, the Arp2/3 complex consists of 8 iso-complexes, as three of its subunits (Arp3, ArpC1, ArpC5) are encoded by two different genes. Here, we used recombinant Arp2/3 with defined composition to assess the activity of CK-666 and CK-869 against iso-complexes. We demonstrate that both inhibitors prevent linear actin filament formation when ArpC1A- or ArpC1B-containing complexes are activated by SPIN90. In contrast, inhibition of actin branching depends on iso-complex composition. Both drugs prevent actin branch formation by complexes containing ArpC1A, but only CK-869 can inhibit ArpC1B-containing complexes. Consistent with this, in bone marrow-derived macrophages which express low levels of ArpC1A, CK-869 but not CK-666, impacted phagocytosis and cell migration. CK-869 also only inhibits Arp3- but not Arp3B-containing iso-complexes. Our findings have important implications for the interpretation of results using CK-666 and CK-869, given that the relative expression levels of ArpC1 and Arp3 isoforms in cells and tissues remains largely unknown.**

**Keywords** Arp2/3 Iso-Complexes; CK-666; CK-869; Inhibition; SPIN90
**Subject Category** Cell Adhesion, Polarity & Cytoskeleton

## Introduction

The Arp2/3 complex is an important evolutionarily conserved nucleator of actin filaments (Machesky et al, 1997; Papalazarou and Machesky, 2021; Welch et al, 1997). The complex regulates the architecture and dynamics of the actin cytoskeleton by initiating the formation of both branched and linear actin filaments (Cao and Way, 2024; Espinoza-Sanchez et al, 2018; Gautreau et al, 2022; Mullins et al, 1998; Wagner et al, 2013). When activated by class I nucleating promoting factors (NPFs), such as WAVE, N-WASP and WASH, the Arp2/3 complex binds to the side of a pre-existing (mother) actin filament and nucleates the formation of a new (daughter) actin branch (Espinoza-Sanchez et al, 2018; Molinie and Gautreau, 2018; Smith et al, 2013). Alternatively, Arp2/3 activation

by SPIN90 (also known as WISH, NCKIPSD, Dip1) inhibits the association of the complex with an existing actin filament and generates a new linear actin filament (Luan et al, 2018; Wagner et al, 2013). Through these two activities, the Arp2/3 complex plays essential roles in various cell processes, including endocytosis, cell migration, regulation of the actin cortex and DNA repair (Cao et al, 2020; Hinze and Boucrot, 2018; Papalazarou and Machesky, 2021; Schrank et al, 2018; Wu et al, 2012).

The Arp2/3 complex consists of seven subunits, including two Actin Related Proteins: Arp2, Arp3 and ArpC1-ArpC5. Upon activation by class I NPFs, the ArpC1–5 subunits interact with the mother actin filament while Arp2 and Arp3 act as a template to nucleate a new daughter filament (Chou et al, 2022; Ding et al, 2022; Fassler et al, 2020). In mammals but not lower eukaryotes, such as yeast and amoeba, Arp3, ArpC1 and ArpC5 exist as two different isoforms that are encoded by separate genes (Balasubramanian et al, 1996; Jay et al, 2000; Millard et al, 2003). In humans, Arp3 and Arp3B, ArpC1A and ArpC1B, ArpC5 and ArpC5L are 91, 67, and 67% identical respectively (Abella et al, 2016). Hence, in mammals, the Arp2/3 complex is not a single species but a group of eight different iso-complexes. Moreover, these iso-complexes have different molecular properties as Arp2/3 complexes containing ArpC1B and ArpC5L are significantly better at stimulating actin assembly than those with ArpC1A and ArpC5 (Abella et al, 2016), while actin networks assembled by Arp3B complexes disassemble faster than those formed by Arp3 complexes (Galloni et al, 2021). Structural analysis reveals that the higher branching efficiency of ArpC5L complexes is mediated by its disordered N-terminus (von Loeffelholz et al, 2020). Not surprisingly, given their different properties, Arp2/3 iso-complexes have different cellular functions (Fassler et al, 2023; Molinie et al, 2019; Roman et al, 2017; Sadhu et al, 2023). The importance of Arp2/3 iso-complexes in cell and tissue homoeostasis is also underscored by the observation that loss of function mutations in human ArpC1B lead to severe inflammation and immunodeficiency (Brigida et al, 2018; Kahr et al, 2017; Kuijpers et al, 2017; Randzavola et al, 2019; Somech et al, 2017; Volpi et al, 2019). It has also recently been reported that loss of ArpC5 expression results in multiple congenital anomalies, recurrent infections, systemic inflammation, and mortality (Nunes-Santos et al, 2023; Sindram et al, 2023).

CK-666 and CK-869 are widely used as chemical inhibitors to probe the function of Arp2/3 complex in biochemical assays, cells and tissues (Nolen et al, 2009; Rotty et al, 2013; Yang et al, 2012). CK-666 binds between Arp2 and Arp3 to block the movement of these two subunits, which is required for activation of the Arp2/3

[1]The Francis Crick Institute, London, UK. [2]Department of Infectious Disease, Imperial College, London, UK. ✉E-mail: luyan.cao@crick.ac.uk; michael.way@crick.ac.uk

complex (Baggett et al, 2012; Hetrick et al, 2013; Rouiller et al, 2008). CK-869 inhibits this activating conformational change by binding to a hydrophobic pocket in Arp3 (Hetrick et al, 2013). CK-869 has also been reported to inhibit microtubule assembly in vitro and cells (Yamagishi et al, 2018). Taking their different properties into consideration, we wondered whether all Arp2/3 iso-complexes can be inhibited by CK-666 and CK-869. In addition, it also remains to be established whether both drugs are equally effective against class 1 NPF- and SPIN90-mediated activation of Arp2/3 iso-complexes. To address these questions, we performed in vitro pyrene and TIRF assays, to examine the impact of CK-666 and CK-869 on the ability of defined recombinant Arp2/3 iso-complexes to nucleate actin when activated by the VCA domain of N-WASP or SPIN90. We found that the ability of CK-666 and CK-869 to inhibit Arp2/3 iso-complexes depends on their subunit composition and activation mechanism. Our observations have important implications for the interpretation of results from experiments using CK-666 and CK-869 to inhibit mammalian Arp2/3 complex both in cells and using purified complexes with unknown isoform composition. They also provide insights into the differences of Arp2/3 iso-complex activation by VCA domains and SPIN90.

# Results and discussion

## CK-869, but not CK-666, fully inhibits Vaccinia-induced actin polymerisation

During their egress from infected cells, newly assembled vaccinia virus virions that have fused with the plasma membrane and remain attached to the cell induce actin polymerisation to enhance their spread into neighbouring uninfected cells (Cudmore et al, 1995; Cudmore et al, 1996; Frischknecht et al, 1999; Leite and Way, 2015). These virions stimulate actin assembly by locally activating Src and Abl family kinases, which results in the recruitment of a signalling network involving the class I NPF, N-WASP that activates Arp2/3 complex-dependent actin polymerisation (Donnelly et al, 2013; Frischknecht et al, 1999; Humphries et al, 2014; Newsome et al, 2004; Newsome et al, 2006; Reeves et al, 2005; Scaplehorn et al, 2002). The resulting actin polymerisation beneath the virion appears as an actin tail when labelled with phalloidin or the branched actin marker cortactin (Fig. 1A). Vaccinia-induced actin tails have been used as a model system to study actin assembly in the cells (Basant and Way, 2022, 2023; Weisswange et al, 2009). We also previously took advantage of vaccinia to demonstrate that Arp2/3 iso-complexes have different cellular activities (Abella et al, 2016; Galloni et al, 2021). We decided to use the same model to examine whether CK-666 and CK-869 differentially impact the ability of the vaccinia to promote branched actin network assembly. Examination of the literature reveals that these drugs are typically used at 100 μM in cell-based assays. Given this, we treated vaccinia-infected HeLa cells 8 h post-infection with 50, 100, 200, and 300 μM CK-666 or CK-869 for 60 min and quantified the ability of the virus to stimulate branched actin assembly using cortactin as a marker (Figs. 1B and EV1). We found that while increasing doses of CK-869 resulted in a near-complete loss of virus-induced actin assembly, CK-666 decreased (~30%) but did not abolish actin polymerisation even at 300 μM (Fig. 1B,C). The actin tails induced by the virus in the presence of CK-666, however, were shorter than in the DMSO control (Fig. 1A,B). The ability of vaccinia

to stimulate actin polymerisation depends on their microtubule transport to the cell periphery and fusion with the plasma membrane (Hollinshead et al, 2001; Leite and Way, 2015; Newsome et al, 2004; Rietdorf et al, 2001). The inhibitory effect of CK-869 on actin assembly, however, was not the consequence of the lack of the virus reaching the plasma membrane as extracellular virions were readily detected on the outside of the cell (Fig. 1B). Our observations on vaccinia-infected HeLa cells demonstrate that CK-666 and CK-869 do not have the same impact on Arp2/3 driven actin assembly.

## CK-666 does not inhibit ArpC1B-containing Arp2/3 complexes

It is not straightforward to deconvolve results from vaccinia-infected cells as the precise levels of the 8 different Arp2/3 iso-complexes, which may not be equally inhibited by CK-666 or CK-869 are not known in HeLa cells. We therefore examined the impact of the two inhibitors on defined recombinant Arp2/3 complexes activated by GST-N-WASP-VCA using in vitro pyrene actin polymerisation assays (Figs. 2 and EV2, EV3). We initially used Arp2/3 iso-complexes containing Arp3 given it is much more abundant than Arp3B in cells and tissues (Galloni et al, 2021; Hein et al, 2015; Kulak et al, 2014). We found both inhibitors suppress activation of ArpC1A containing Arp2/3 complexes but only CK-869 and not CK-666 can inhibit those with ArpC1B (Fig. 2A). A similar difference in inhibition was seen in complexes with ArpC5 or ArpC5L (Fig. EV3). The lack of inhibition of ArpC1B complexes by CK-666 was unexpected, given that the drug binds between Arp2 and Arp3 to block their movement during Arp2/3 complex activation (Baggett et al, 2012; Hetrick et al, 2013).

In vitro pyrene actin polymerisation assays are an indirect measurement of bulk actin assembly, so they provide no information at the level of individual filaments. To directly quantify actin branch initiation efficiency, we therefore performed TIRF assays where the generation of new branches is observed over time by incubating fluorescent actin filaments with G-actin labelled with a different fluorophore (Risca et al, 2012). We found that 100 μM CK-869 dramatically reduces the branching rate of ArpC1A/C5, ArpC1A/C5L, ArpC1B/C5, and ArpC1B/C5L containing complexes (Fig. 2B). In contrast, 100 μM CK-666 was only able to inhibit ArpC1A containing Arp2/3 complexes.

Previous in vitro analysis determined that the half-maximal inhibitory concentration value (IC$_{50}$) of CK-666 for human and bovine Arp2/3 purified from platelets and thymus was 4 and 17 μM respectively, while it was 11 μM for CK-869 on bovine Arp2/3 (Nolen et al, 2009). As these Arp2/3 complexes are purified from natural sources they are likely to contain different iso-complexes. Given this and our in vitro observations, we determined the IC$_{50}$ for CK-666 and CK-869 on defined Arp2/3 iso-complexes activated by VCA (Fig. 2C). We were unable to determine an IC$_{50}$ for ArpC1B/C5L but obtained a value of 19.9 μM for CK-666 with ArpC1A/C5L containing complexes. The lower IC$_{50}$ values we obtained for CK-869 clearly demonstrate that the drug is a significantly better inhibitor than CK-666. Moreover, in an ArpC5L background, CK-869 is more effective against complexes containing ArpC1A (IC$_{50}$ 0.86 μM) than those with ArpC1B (IC$_{50}$ 3.55 μM) (Fig. 2C). Although the ArpC5/ArpC5L isoforms are not differentially impacted by either CK-666 or CK-869 (Fig. 2B), the latter appears slightly more effective against ArpC5 containing complexes

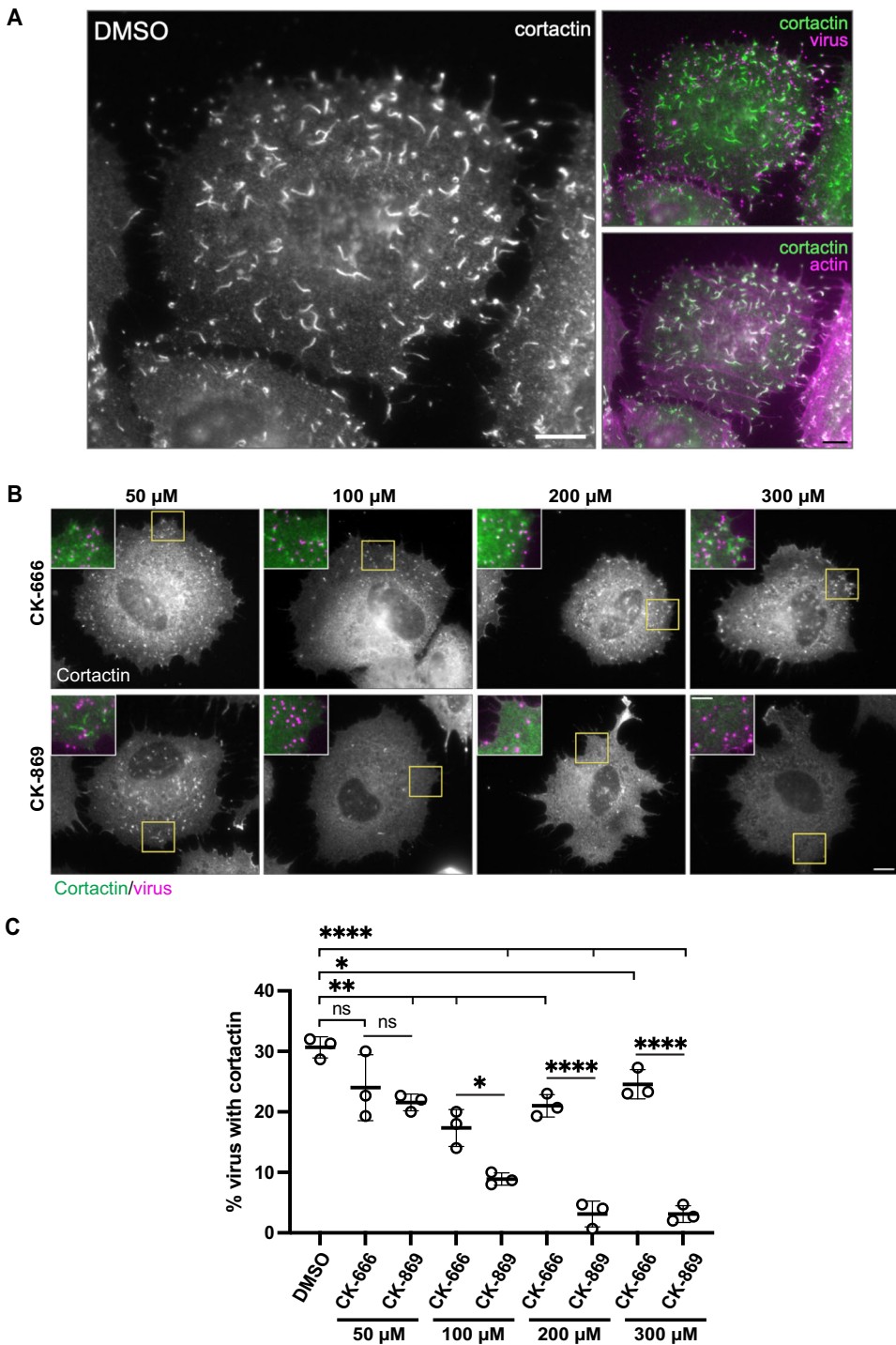

**Figure 1. CK-869, but not CK-666 fully inhibits vaccinia-induced actin polymerisation.**

(A) Representative immunofluorescence image of a HeLa cell infected with vaccinia virus at 9 h post-infection and labelled with an antibody against cortactin and stained with phalloidin to visualise F-actin. Extracellular viruses attached to the plasma membrane are labelled with an antibody against the viral protein B5. Scale bar = 10 μm. (B) Representative images showing localisation of cortactin in HeLa cells infected with vaccinia at 9 h post-infection after 1 h treatment with CK-666 or CK-869 at the specified concentrations. Scale bar = 10 μm. The inset colour images show 2x magnifications of the yellow boxed regions with cortactin (green) and extracellular virus (magenta). (C) The graph shows quantification of the mean number of extracellular viruses co-localising with cortactin from three independent repeats with the error bar representing standard deviation. 10 cells and 150 virus particles were analysed in each experiment. A two-tailed unpaired *t*-test was used to analyse the statistical significance between different drugs at the same concentration as well as between the drug and DMSO control. ns not significant. *p value <0.05. **p value <0.01. ****p value <0.0001. Source data are available online for this figure.

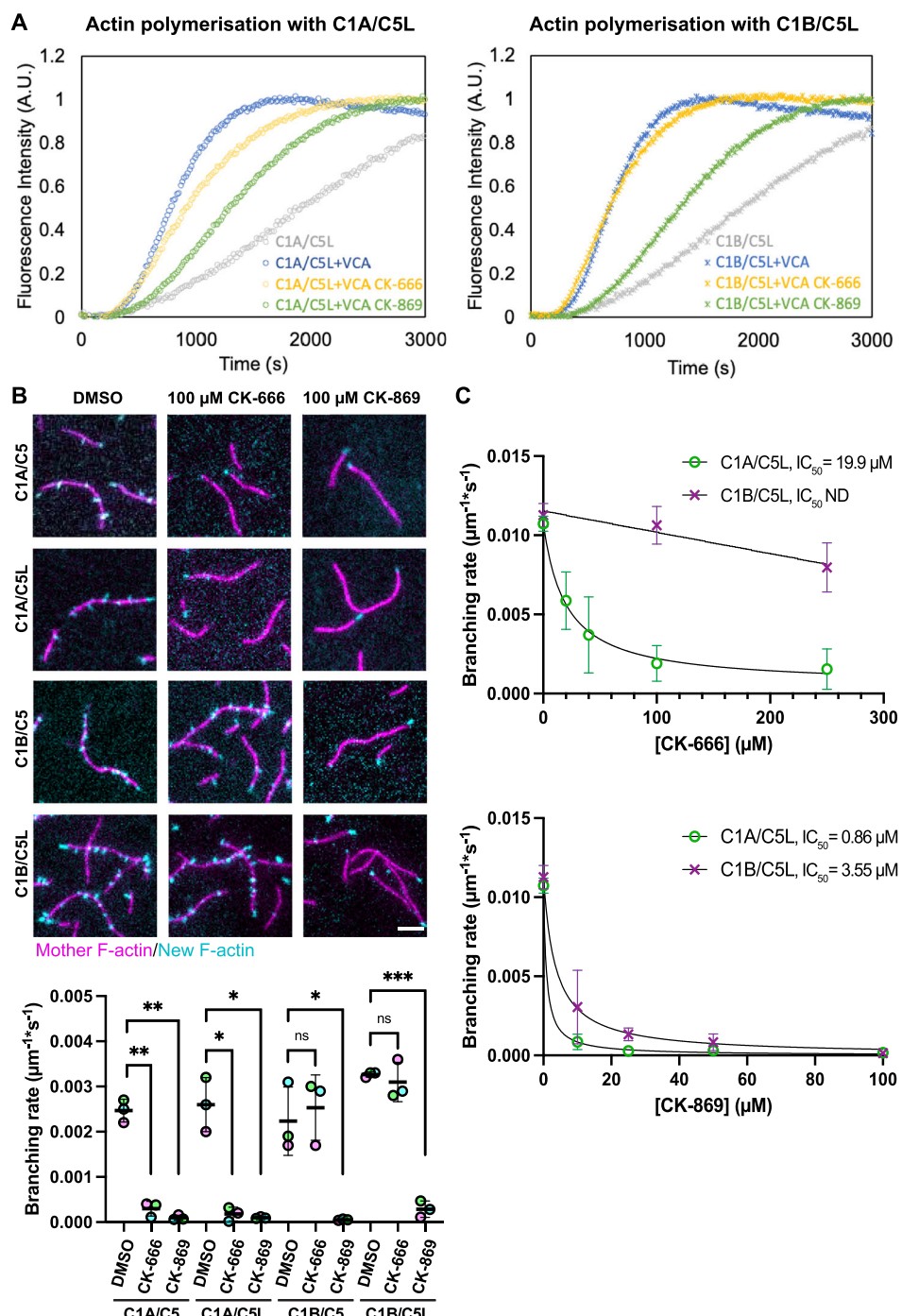

**Figure 2. CK-666 does not inhibit ArpC1B-containing Arp2/3 complexes.**

(A) Representative plots showing the polymerisation of pyrene-labelled actin (Fluorescence Intensity) when Arp2/3 iso-complexes containing ArpC1A/C5L (left) or ArpC1B/C5L (right) are activated by GST-N-WASP-VCA in the absence (blue) or presence of 100 μM CK-666 (yellow) or CK-869 (green). (B) Representative TIRF images showing mother filaments (magenta) and new daughter branches (cyan) 2 min after Arp2/3 iso-complexes containing ArpC1A/C5, ArpC1A/C5L, ArpC1B/C5, or ArpC1B/C5L were activated by GST-N-WASP-VCA in the absence (DMSO) or presence of 100 μM CK-666 or CK-869. Scale bar = 5 μm. The graph shows the quantification of the mean branching rate for each condition from three independent experiments with the error bar representing the standard deviation. Two-tailed paired t-test has been applied to analyse the statistical significance. *p value <0.05. **p <0.01. ***p value <0.001. ns not significant. (C) The branching rate of the Arp2/3 complex containing ArpC1A/C5L and ArpC1B/C5L were measured at the indicated concentrations of CK-666 (top) and CK-869 (bottom). The data was fitted with equation $Y = \text{Bottom} + (\text{Top-Bottom})/(1 + (X/\text{IC}_{50}))$ to calculate the half-maximal inhibitory concentration of the drugs ($\text{IC}_{50}$), which is indicated (ND not determined). The points and error bars represent the mean and the standard deviation from at least two independent measurements. Source data are available online for this figure.

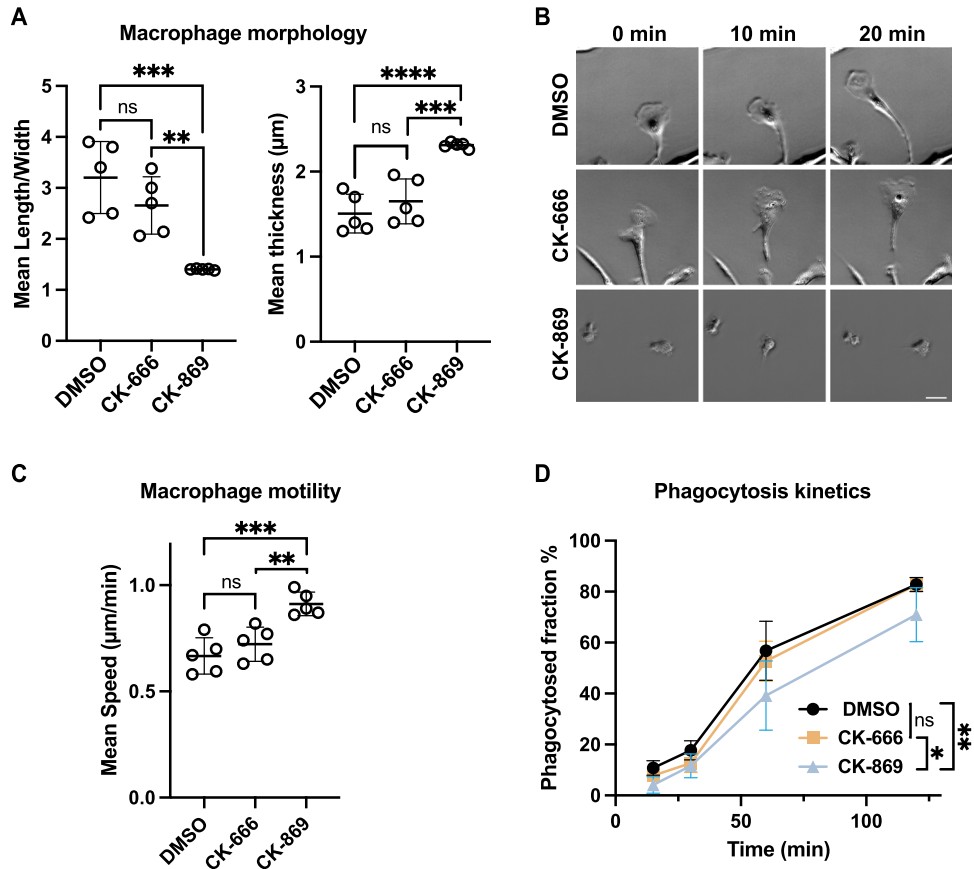

**Figure 3. CK-666 has no adverse impact on murine bone marrow-derived macrophages.**

(A) Treatment of murine bone marrow-derived macrophages (BMDM) with 100 μM CK-869 but not DMSO or CK-666 induces cell rounding. The graphs show the mean ratio of length and width as well as cell thickness from five independent experiments with error bars representing the standard deviation. In each experimental condition, at least 30 cells were randomly chosen and analysed. A two-tailed unpaired *t*-test was used to analyse the statistical significance. (B) Representative phase images of BMDM treated with DMSO, CK-666, or CK-869 over 20 min. Scale bar = 20 μm. (C) Quantification of the mean migration speed of BMDM treated with DMSO, CK-666, or CK-869 from five independent experiments with error bars representing the standard deviation. In each experimental condition, at least 30 cells were randomly chosen and analysed. A two-tailed, unpaired student *t*-test was used to analyse the statistical significance. (D) The graph shows the quantification of the phagocytosis efficiency of zymosan by murine bone marrow-derived macrophages treated with DMSO, CK-666 or CK-869 over 120 min. In each experimental condition, $10^4$ cells were analysed using flow cytometry. Each point represents the mean of three independent technical repeats and error bars represent the standard deviation. A two-way ANOVA test has been applied to analyse the statistical significance. *$p$ value <0.05. **$p$ value <0.01. ***$p$ value <0.001. **** means $p$ value <0.0001. ns not significant. Source data are available online for this figure.

(Fig. EV3B). Our results clearly demonstrate that CK-666 inhibits ArpC1A but not ArpC1B-containing Arp2/3 complexes from generating actin branches, whereas CK-869 inhibits both ArpC1 iso-complexes. The results from our in vitro assays offer an explanation for why CK-666 does not fully suppress vaccinia-induced actin polymerisation in HeLa cells, which express ~2.3 times as much ArpC1B as ArpC1A (Abella et al, 2016).

## CK-666 has no impact on macrophage phagocytosis and motility

Single-cell transcriptomics, tissue-specific mRNA expression and proteomic analysis indicate that ArpC1B is highly expressed in immune cells compared to ArpC1A (Kahr et al, 2017; Karlsson et al, 2021; Kim et al, 2014; Wu et al, 2009). Given this, we predicted that in contrast to CK-869, CK-666 will have little or no effect on Arp2/3 complex-dependent activities in immune cells. After incubating

murine bone marrow-derived macrophages with 100 μM of Arp2/3 inhibitor, we observed that the morphology of macrophages treated with CK-869 changed significantly, with cells rounding up to become less spread (Fig. 3A,B; Movie EV1). In contrast, CK-666-treated macrophages were indistinguishable from DMSO-treated controls. In addition, treatment with CK-666 had no impact on macrophage motility (Fig. 3C; Movie EV1). CK-869, on the other hand, unexpectedly increases macrophage movement. However, a similar increase in motility has been observed in macrophages that contain no functional Arp2/3 complex because they lack ArpC2 (Rotty et al, 2017). This study demonstrated that this effect was dependent on myosin II and suggested that the increased motility was due to weakened cell adhesion and enhanced cell contractility. Rotty et al also found that the FcR phagocytic capability of CK-666-treated macrophages was similar to DMSO-treated controls. Extending this observation, we found that CK-869 but not CK-666 decreased the ability of macrophages to phagocytose zymosan compared to

DMSO-treated controls (Figs. 3D and EV4). Our observations in macrophages suggest that the ability of CK-666 to inhibit Arp2/3-dependent processes will depend on the relative expression of ArpC1A compared to ArpC1B, which will vary across cell types and tissues.

## CK-666 and CK-869 do not disrupt the Arp2/3 complex binding to F-actin or VCA

Structural data indicate that neither CK-666 nor CK-869 binds directly to the ArpC1 subunit (Nolen et al, 2009). The divergent effects of CK-666 on ArpC1A and ArpC1B within the Arp2/3 complex remains enigmatic. A possible explanation may be that the drugs affect the interaction between Arp2/3 iso-complexes and actin filaments indirectly. To investigate this, we performed actin filament co-sedimentation assays using ArpC1A and ArpC1B-containing complexes in the absence of VCA-mediated activation. We found that regardless of the presence of drugs, comparable quantities of ArpC1A and ArpC1B-containing Arp2/3 complexes co-pelleted with 3 or 15 μM actin filaments (Fig. EV5A). This indicates that CK-666 and CK-869 do not impair the interaction between the Arp2/3 complex and actin filaments, in the context of different ArpC1 isoforms. However, when assays were performed with 3 μM F-actin, the level of ArpC1B/C5L co-pelleting with actin is a nearly twofold greater than complexes containing ArpC1A/C5L (Fig. EV5A). This may, in part, explain why the Arp2/3 complex with ArpC1B/C5L is a more efficient branch generator than those with ArpC1A/C5L (Fig. 2B) (Abella et al, 2016).

Previous observations indicate that CK-666 and CK-869 do not impact VCA binding (Hetrick et al, 2013). However, in this study, the Arp2/3 complex will be a mixture of iso-complexes as it was isolated from the bovine brain (Abella et al, 2016). We, therefore, performed GST-N-WASP-VCA pulldown assays to capture ArpC1A and ArpC1B-containing complexes in the presence or absence of Arp2/3 inhibitors to determine if there are iso-complex-specific differences. We found that the presence of either drug did not alter the quantity of ArpC1A- and ArpC1B-containing Arp2/3 complexes bound to GST-N-WASP-VCA (Fig. EV5B). These results are consistent with previous observations, irrespective of the presence of distinct ArpC1 isoforms (Hetrick et al, 2013). It also suggests that inhibited and uninhibited Arp2/3 iso-complexes will compete for VCA binding in cells, which has important implications depending on the relative levels of ArpC1A and ArpC1B and which inhibitor is used. It would also explain why CK-666 inhibited ArpC1A complexes, which represent about a third of the total Arp2/3 in our HeLa cells (Abella et al, 2016), decreased virus-induced actin polymerisation by ~30% (Fig. 1B,C), even though Arp2/3 is in excess of N-WASP (Hein et al, 2015; Kulak et al, 2014).

## CK-666 inhibits activation of all Arp2/3 iso-complexes by SPIN90

As CK-666 inhibits ArpC1A, but not ArpC1B, containing Arp2/3 complexes from generating actin branches, we investigated whether the same is true when Arp2/3 is activated to form linear actin filaments by SPIN90. Pyrene assays demonstrate that CK-666 and CK-869 inhibit SPIN90-dependent actin polymerisation with a similar $IC_{50}$ irrespective of the ArpC1 isoform (Figs. 4A and EV6). We also observed by TIRF microscopy that both drugs inhibit actin filament formation by ArpC1A and ArpC1B containing Arp2/3

complexes activated by SPIN90, but only CK-869 completely blocked actin assembly (Fig. 4B).

Neither drug inhibited the interaction of Arp2/3 with SPIN90 in GST pull-down assays (Fig. 4C). However, unexpectedly CK-869 promotes the binding of Arp2/3 to SPIN90, with the interaction with ArpC1A and ArpC1B containing complexes increasing ~15 and fivefold, respectively. There was also a slight increase in binding in the presence of CK-666. A possible explanation is that CK-869 blocks the Arp2/3 complex in an inactive state, which favours binding to SPIN90 but prevents further conformational changes required for full activation which would subsequently weaken the association with SPIN90. This would imply that inactive Arp2/3 has a higher affinity for SPIN90 than the active complex, as is observed for VCA but not cortactin NtA (Liu et al, 2024; Zimmet et al, 2020). This may explain why the amount of Arp2/3 pulled down with GST-SPIN90 is reduced in the presence of GST-N-WASP-VCA (Cao et al, 2023).

## CK-869 cannot inhibit Arp3B-containing complexes

Our results thus far demonstrate that CK-869 is a better inhibitor of Arp3-containing complexes than CK-666. To explore whether the same is also true for Arp3B-containing complexes, we examined the impact of both inhibitors on the ability of Arp3B/ArpC1B/ArpC5L containing Arp2/3 complex to nucleate actin filaments. Quantification of actin assembly in TIRF assays reveals that the branching rates induced by GST-N-WASP-VCA in the presence of CK-666 were indistinguishable from DMSO-treated controls (Fig. 5A). Unexpectedly, CK-869 no longer inhibits the Arp3B-containing complex, which contrasts with our earlier observations using complexes with Arp3 (Fig. 2). Moreover, as observed with Arp3-containing complexes (Fig. 4), CK-666 inhibits the ability of SPIN90 to stimulate linear actin filament assembly by Arp3B-containing complexes (Fig. 5B). In sharp contrast, Arp3B-containing complexes efficiently generate linear actin filaments in the presence of CK-869. Our in vitro data demonstrate that CK-869 cannot inhibit Arp3B-containing complexes, even up to 250 μM in pyrene actin assembly assays (Fig. 5C).

Previous structural analysis reveals that CK-666 interacts mostly with Arp2 but forms two hydrophobic interactions with leucine 117 and threonine 119 of Arp3 (Baggett et al, 2012). These two residues are conserved in Arp3B, suggesting that the binding of CK-666 will be similar, which is consistent with our in vitro data. In contrast, several residues in the hydrophobic CK-869 binding pocket of Arp3 differ from those in Arp3B (Hetrick et al, 2013) (Fig. 5D). Notably, the leucine 126 in Arp3B may have direct clashes with CK-869 and preclude its binding leading to a loss of inhibition. A similar lack of inhibition due to non-conserved residues in the CK-869 binding pocket is also observed in Arp2/3 from budding and fission yeast (Nolen et al, 2009).

Using purified components and in vitro assays we have demonstrated that CK-666 can inhibit the formation of SPIN90-induced linear actin filaments by all Arp2/3 iso-complexes. However, CK-666 fails to suppress the initiation of actin branch formation by Arp2/3 complexes containing ArpC1B. In contrast, CK-869 prevents Arp3 but not Arp3B-containing complexes from generating both actin branches and linear actin filaments. Our results provide further evidence that Arp2/3 iso-complexes have different molecular properties. This observation also aligns with

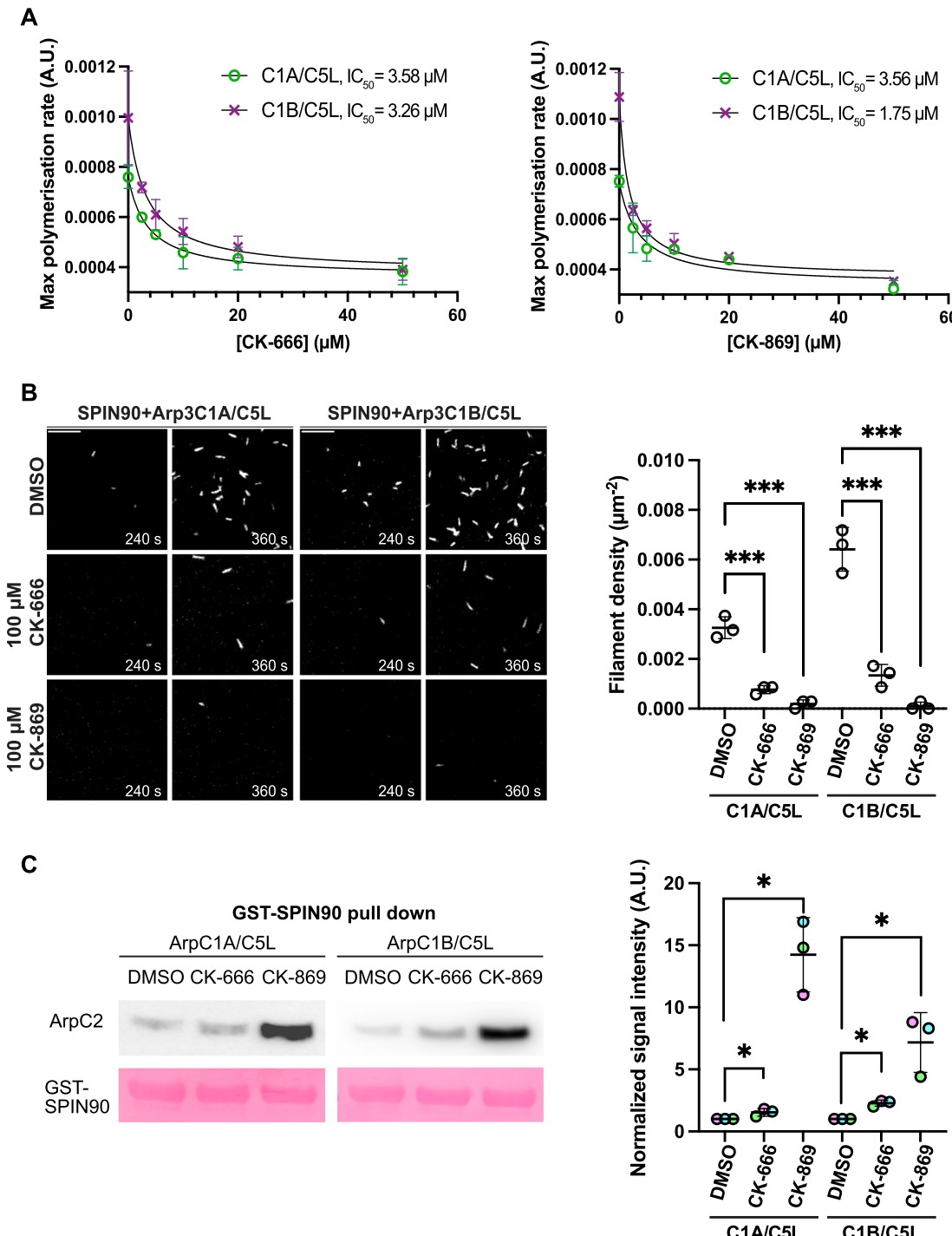

**Figure 4.  The impact of CK-666 and CK-869 on SPIN90 activation of Arp2/3 complexes.**

(**A**) The maximum rate of pyrene actin polymerisation stimulated by SPIN90-Arp2/3 iso-complexes containing ArpC1A/C5L and ArpC1B/C5L in the presence of the indicated concentrations of CK-666 (top) and CK-869 (bottom). The data was fitted with equation $Y = Bottom + (Top-Bottom)/(1 + (X/IC_{50}))$ to calculate the half-maximal inhibitory concentration of the drugs ($IC_{50}$), which is indicated. The points and error bars represent the mean and the standard deviation of at least two independent measurements. (**B**) Representative TIRF images of actin filament assembly induced by Arp2/3 complexes containing ArpC1A/C5L or ArpC1B/C5L after activation by SPIN90-Cter in the absence (DMSO) or presence of 100 μM CK-666 or CK-869 at the indicated times. Scale bar = 10 μm. The graph shows the quantification of the mean filament density at 240 s from three independent experiments, with the error bar representing the standard deviation. Two-tailed unpaired $t$-test was used to analyse the statistical significance and ***$p$ value <0.001. (**C**) The left panels show immunoblots of GST-SPIN90-Cter (Ponceau - red) pulldown assays of the indicated Arp2/3 iso-complexes (detected by ArpC2 antibody) in the absence (DMSO) or presence of 100 μM CK-666 or CK-869. The right panel shows the quantification of the mean bound Arp2/3 complex normalised to the DMSO control for three independent pulldown assays with the error bar representing standard deviation. Two-tailed paired $t$-test was used to analyse the statistical significance. ns not significant. *$p$ value <0.5. Source data are available online for this figure.

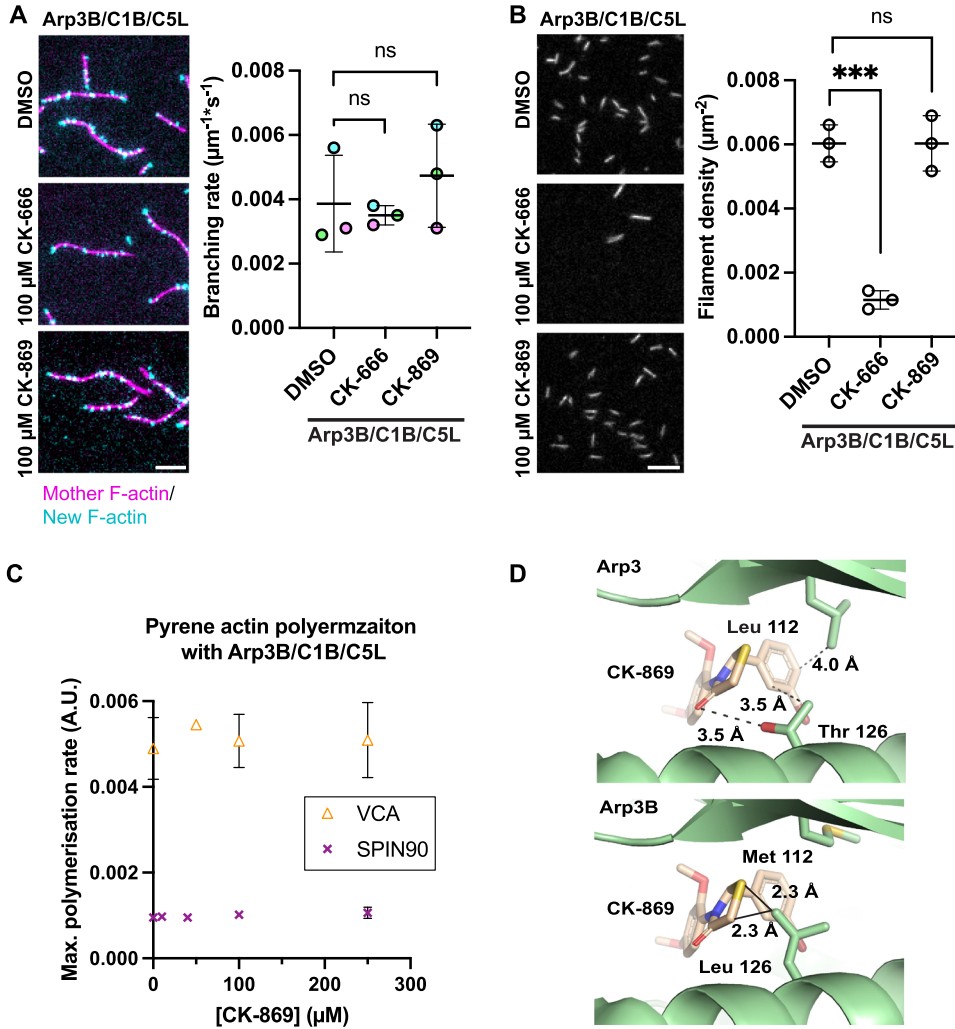

**Figure 5. Impacts of drugs on Arp3B-containing complexes.**

(A) Representative TIRF images showing mother filaments (magenta) and newly polymerised daughter actin branches (cyan) 2 min after Arp2/3 complex containing Arp3B/ArpC1B/C5L was activated by GST-N-WASP-VCA in the absence (DMSO) or presence of 100 μM CK-666 or CK-869. Scale bar = 5 μm. The graph shows the quantification of the mean branching rate for each condition from three independent experiments with the error bar representing the standard deviation. A two-tailed paired *t*-test was used to analyse the statistical significance and ns not significant. (B) Representative TIRF images of actin filament assembly induced by Arp2/3 complex containing Arp3B/ArpC1B/C5L after activation by SPIN90-Cter in the absence (DMSO) or presence of 100 μM CK-666 or CK-869 at the indicated times. Scale bar = 10 μm. The graph shows the quantification of the mean filament density at 240 s from three independent experiments with the error bar representing the standard deviation. A two-tailed unpaired *t*-test was used to analyse the statistical significance. ns not significant. *** presents *p* value <0.001. (C) The maximum rate of pyrene actin polymerisation stimulated by Arp2/3 complex containing Arp3B/ArpC1B/C5L activated by either GST-N-WASP-VCA or SPIN90-Cter at the indicated CK-869 concentration. The points and error bars represent the mean and the standard deviation of at least two independent measurements. (D) The top image shows that in the Arp2/3 complex (PDB: 3ULE), CK-869 has interaction with Leu 112 and Thr 126 of Arp3. The bottom image shows that in Arp3B, Leu 112, and Thr 126 are replaced by Met 112 and Leu 126, the latter of which is predicted to have clashes with CK-869. Source data are available online for this figure.

our cellular findings, where CK-869 exhibited greater efficacy in inhibiting vaccinia-induced actin assembly compared to CK-666. Examination of the literature reveals that most studies typically treat cells with CK-666 rather than CK-869, presumably because the latter is more toxic as it will inhibit Arp2/3 complexes containing Arp3, which are much more abundant than those with Arp3B (Galloni et al, 2021; Hein et al, 2015; Kulak et al, 2014).

Our data demonstrate that both drugs have high affinity for Arp2/3 iso-complexes containing Arp3 (Fig. 4A). However, CK-666 and CK-869 have a different impact on Arp2/3 iso-complexes, depending on whether they are activated by VCA or SPIN90. Both

drugs can effectively inhibit Arp2/3-SPIN90 generating actin filaments, while simultaneously enhancing, to varying degrees, the binding of Arp2/3 complexes to SPIN90. This implies that SPIN90 may induce a relatively weak activation of Arp2/3 complexes, which is more susceptible to inhibition.

The observation that CK-666 has no discernible effect on the binding of Arp2/3 to either VCA or mother filaments indicates that CK-666 primarily disrupts the conformational changes of Arp2 and Arp3 to prevent complex activation as suggested by the structural data (Baggett et al, 2012). Nevertheless, ArpC1B but not ArpC1A containing complexes are able to adopt an activated conformation

to generate actin branches on binding VCA even though they are bound to CK-666. This would suggest that the energy barrier between inactive and active conformations is lower for ArpC1B containing Arp2/3 complexes than those with ArpC1A. It is also possible that the conformation change required for activation displaces CK-666 from Arp2/3 iso-complexes containing ArpC1B. Structural analysis of Arp2/3 iso-complexes in transitional states of complex activation is required to fully understand their different behaviour.

The pivotal role of Arp2/3 complexes in many cellular processes that can become dysregulated in pathological conditions, highlights the ongoing need to screen for additional Arp2/3 inhibitors. Suppressing cancer cell migration and tumour metastasis, two ArpC2 binding inhibitors, were recently identified by revisiting FDA-approved drugs (Choi et al, 2019; Yoon et al, 2019). Several CK-666 structural analogues have been reported to have an inhibitory effect on the Arp2/3 complex and two of them have better in vivo efficiency compared with CK-666 (Fokin et al, 2022). Given our results, it is possible that these drugs have different inhibitory effects on Arp2/3 iso-complexes. In summary, our results have important implications for the interpretation of experiments with CK-666 and CK-869 but also illuminate the potential utility of these inhibitors in exploring the cellular functions of specific Arp2/3 iso-complexes.

# Methods

## Vaccinia virus infection and drug treatment

HeLa cells plated on fibronectin-coated coverslips were infected with Western Reserve Vaccinia virus in serum-free minimum essential medium (MEM) at a multiplicity of infection (MOI) = 2 (Frischknecht et al, 1999). After 1 h at 37 °C, the serum-free MEM was removed and replaced with complete MEM. At 8 h post-infection, media was replaced with complete MEM containing CK-666 (Sigma, SML0006-5MG) or CK-869 (Sigma, C9124-5MG) or DMSO at desired concentrations for 1 h prior to fixation.

## Immunofluorescence and actin tail quantification

As previously described (Basant and Way, 2022), cells were fixed with 4% paraformaldehyde in PBS for 10 min, blocked in cytoskeletal buffer (1 mM MES, 15 mM NaCl, 0.5 mM EGTA, 0.5 mM MgCl$_2$, and 0.5 mM glucose, pH 6.1) containing 2% (vol/vol) foetal calf serum and 1% (wt/vol) BSA for 30 min. To visualise extracellular vaccinia virus particles, prior to permeabilization with detergent, the cells were stained with a monoclonal antibody against B5 (19C2, rat, 1:1000 (Schmelz et al, 1994)) followed by an Alexa Fluor 647 anti-rat secondary antibody (Invitrogen; 1:1000 in blocking buffer). Cells were then permeabilized with 0.1% Triton-X/PBS for 3 min. Branched actin was labelled with cortactin monoclonal antibody (Millipore (p80/85); 1:400), followed by Alexa Fluor 488 conjugated secondary antibody (Invitrogen; 1:1000 in blocking buffer). Actin filaments were labelled with Alexa Fluor 568 phalloidin (Invitrogen; 1:500). Coverslips were mounted on glass slides using Mowiol (Sigma) and imaged on a Zeiss Axioplan2 microscope equipped with a 63x/1.4 NA. Plan-Achromat objective and a Photometrics Cool Snap HQ cooled charge-coupled device

camera. The microscope was controlled with MetaMorph 7.8.13.0 software. Quantification of branched actin induced by vaccinia virus was performed using two-colour fixed cell images where cortactin and extracellular virus were labelled. Ten cells were analysed per condition in three independent experiments. Fifteen isolated extracellular virus particles were blindly selected in each image and the presence of actin and cortactin in the virus was determined in the corresponding channels.

## Bone marrow-derived macrophage migration and phagocytosis

All animal work was authorised by UK Home Office project license P7E080263 and personal licences following the approval by the Animal Welfare and Ethical Review Body of The Francis Crick Institute. Murine bone marrow was extracted from the tibia and femur of C57BL/6Jax mice and differentiated in vitro for 7 days in macrophage conditional medium (RPMI supplemented with 30% L929 conditioned medium, 20% FBS and 1% P/S to obtain bone marrow-derived macrophages (BMDM).

To examine the effects of different inhibitors on cell morphology and migration, BMDM were seeded in 24-well glass-bottomed plates (Cellvis) at $1 \times 10^4$ cells/well in 500 μL of macrophage conditional medium. Cells were imaged and analyzed using the Livecyte system (Phasefocus) every 10 min for 24 h at 10x with two fields of view per well. The imaging chamber was maintained at 37 °C with 5% CO$_2$. Meanwhile, cells were seeded at $2 \times 10^4$ cells/well in Glass-bottomed 8-well μ-Slides (ibidi) and imaged using an inverted widefield Nikon Ti2 Eclipse long-term time-lapse system with an LED illumination system. The imaging chamber was maintained at 37 °C with 5% CO$_2$. A Nikon 20× Ph2 planar (plan) apochromatic (apo) (0.75 N.A.) air objective was used to acquire images at a single $z$-slice every 10 min for 5 h with two fields of view per well. Livecyte automatically quantified the motility and the morphology of macrophages by measuring their moving speed as well as their thickness and their length/width ratio. For each condition, more than 30 cells were quantified. The experiments were repeated five times independently.

To assess the effects of different inhibitors on phagocytosis, BMDM were seeded in 96-well V-bottom plates (Corning) at $1 \times 10^5$ cells/well in 100 μL macrophage conditional medium supplemented with different inhibitors or DMSO as indicated. BMDM were incubated with 25 μg/mL pHrodo zymosan bio-particles (Invitrogen) for up to 120 min at 37 °C, 5% CO$_2$. Cells were then washed with PBS and stained with an antibody cocktail (Live/Dead Fixable Violet Dead Cell Stain (1:500, Invitrogen), CD11b-BV711 (1:100, BioLegend), F4/80-APC (1:100, eBiosciences)) to identify the live BMDM. Samples were then assessed by flow cytometry and the phagocytic fraction indicated the % of (BMDM internalised the zymosan bio-particles)/(Total BMDM).

## Protein purification and preparation

GST-tagged human N-WASP-VCA (392 – 505, UniProt O00401), human SPIN90 C-terminus (267 - 715, UniProt Q9NZQ3-3), were purified as previously reported (Cao et al, 2023). Recombinant human Arp2/3 iso-complexes with defined composition (Uniprot P61160, P61158, Q9P1U1, Q92747, O15143, O15144, O15145, P59998, Q9H9F9, and Q98PX5) were purified as previously reported (Baldauf

et al, 2023). Skeletal muscle alpha-actin was purified from rabbit muscle acetone powder following the published protocol (Spudich and Watt, 1971). Actin was fluorescently labelled on the surface lysine 328, using Alexa-488 or Alexa-594 succinimidyl ester (Life Technologies) as previously reported (Romet-Lemonne et al, 2018). Pyrenyl-actin was made by labelling actin with N(1-pyrene)-iodoacetamide (Thermo Fisher Scientific) (Cooper et al, 1983). To mimic the use of drugs in cells, Arp2/3 iso-complexes were incubated with $100\,\mu M$ CK-666 or CK-869 (Sigma, Bio-Techne), or equivalent amount of DMSO at room temperature for 1 h before being used in in vitro assays.

## Pyrene actin polymerisation assay

Actin polymerisation induced by Arp2/3 and NPF I is very fast. To have a good resolution, 2 nM Arp2/3 iso-complexes, 5 nM GST-N-WASP-VCA and $2\,\mu M$ G-actin (5% pyrene labelled) was mixed and measured in a Safas Xenius fluorimeter at room temperature (von Loeffelholz et al, 2020). The negative control experiment was done without GST-N-WASP-VCA. For experiments with SPIN90, the same concentration of Arp2/3 iso-complexes was used together with 100 nM SPIN90-Cter and $2.5\,\mu M$ G-actin (5% pyrene labelled). The negative control for these experiments lacked SPIN90-Cter. All actin assembly assays above were performed in a buffer containing 5 mM Tris-HCl pH 7.0, 50 mM KCl, 0.2 mM ATP, 1 mM $MgCl_2$, 1 mM EGTA, and 1 mM DTT in addition to $100\,\mu M$ CK-666 or CK-869 or equivalent amount of DMSO. To quantify $IC_{50}$ values, Arp2/3 iso-complexes were incubated with DMSO or different drug concentrations for 1 h. Then 2 nM drug-treated Arp2/3 iso-complexes were mixed with 10 nM VCA or 400 nM SPIN90-Cter together with $2.5\,\mu M$ G-actin (5% pyrene labelled) in the presence of the same concentration of drugs. The experimental buffer contains 5 mM Tris-HCl pH 7.0, 50 mM KCl, 1 mM $MgCl_2$, 0.2 mM EGTA, 0.2 mM ATP, 10 mM DTT, and 1 mM DABCO at room temperature. The maximum polymerisation rate of each pyrene curve was measured and plotted versus the defined drug concentration.

## TIRF assays

Deep UV-treated coverslips were passivated by mPEG silane overnight. The coverslips were rinsed thoroughly with ethanol and water. Before experiments, dried coverslips were stuck to the glass slides with 3-mm-separated parafilm bands to create flow chambers. The experiments were performed in TIRF buffer: 5 mM Tris-HCl pH 7.0, 50 mM KCl, 1 mM $MgCl_2$, 0.2 mM EGTA, 0.2 mM ATP, 10 mM DTT, 1 mM DABCO, 0.1% BSA, and 0.3% Methylcellulose 4000 cP at 25 °C.

Actin branching assays were performed as previously described (Cao et al, 2020). To quantify the branching efficiency, $2.5\,\mu M$ G-actin (15% Alexa-488 labelled) was pre-incubated in an experimental buffer at room temperature for 1 h to form fluorescent actin filaments. During the experiment, 25 nM pre-polymerised actin (15% Alexa-488 labelled) was mixed with 25 nM GST-N-WASP-VCA, $0.5\,\mu M$ profilin, 2 nM of 1 h inhibitor-treated Arp2/3 in addition to defined concentration of CK-666 or CK-869 (final concentration) or equivalent amount of DMSO. G-actin ($0.5\,\mu M$, 15% Alexa-568 labelled) was added to the system at the last moment. Then the mixture was immediately flowed into the chamber for imaging. The moment when G-actin was added into the system was recorded as time 0. Branch generation was observed over time. The number of branches generated per $\mu m$ of actin filaments per minute was quantified as the branching rate. The quantified branching rates versus defined drug concentrations were plotted to determine the $IC_{50}$. To study the actin nucleation by SPIN90, the 25 nM GST-N-WASP-VCA was replaced with 200 nM SPIN90-Cter and 20 nM of inhibitor-treated Arp2/3 were used in the same protocol as above.

The actin branching rate in the TIRF assay was quantified by counting the number of generated branches per $\mu m$ length of actin mother filament per second using Fiji software (Schindelin et al, 2012). In each condition, the ten longest mother filaments were chosen to analyse. Three independent repeats were performed. Actin nucleation by SPIN90-Arp2/3 complex in TIRF assay was quantified by measuring the filament density of a randomly picked field of view ($59\,\mu m \times 59\,\mu m$) at 240 s. Three independent repeats were performed.

## GST pull-down assays

Prior to performing pull-down assays, the Glutathione Sepharose™ 4B resin (GE Healthcare) was incubated with 5% BSA for 5 min and washed three times with GST pull-down buffer containing 50 mM Tris-HCl pH 7.0, 1 mM $MgCl_2$, 0.2 mM EGTA, 0.2 mM ATP, 1 mM DTT, 50 mM KCl, and 5% glycerol. Glutathione Sepharose 4B beads were mixed with $50\,\mu L$ of $5\,\mu M$ GST-SPIN90-Cter or GST-N-WASP-VCA, 200 nM drug-treated Arp2/3 in addition to $100\,\mu M$ inhibitor or equivalent amount of DMSO. After 1 h incubation at room temperature, the resin was then washed three times with $300\,\mu L$ of GST pull-down buffer. The proteins bound to the resin were eluted with $50\,\mu L$ of 50 mM GSH (reduced glutathione). The sample was analysed by SDS-PAGE followed by western blot using Anti-ArpC2 (Sigma) to detect Arp2/3 complexes. Amersham imager AI600 was used to develop the western blot results. The amount of GST-tagged protein eluted from the resin was detected by ponceau. The negative control experiment was done in the same way except $50\,\mu L$ of $5\,\mu M$ Glutathione Sepharose™ 4B attached GST was used instead of GST-SPIN90-Cter or GST-N-WASP-VCA.

## F-Actin co-sedimentation assays

G-actin was incubated in F-buffer (5 mM Tris pH 7.0, 100 mM KCl, 0.2 mM ATP, 1 mM $MgCl_2$, and 1 mM DTT) at room temperature for 1 h before 3 or $15\,\mu M$ of the polymerised actin filaments were mixed with $4\,\mu M$ inhibitor-treated Arp2/3 iso-complexes in F-buffer in the presence of $100\,\mu M$ CK-666 or CK-869 or equivalent amount of DMSO at room temperature for 1 h. The supernatant and pellet were then separated by ultracentrifuge at 70,000 rpm at 25 °C for 30 min. The pellet was resuspended with $50\,\mu L$ of F-buffer and the same amount ($20\,\mu L$) of supernatant and pellet was loaded on an SDS-page. The negative control only has Arp2/3 iso-complexes without F-actin. Fiji software was used to quantify the Coomassie blue gels (Schindelin et al, 2012).

## Protein structure analysis

Pymol is used to visualise and analyse the structure of drug bound Arp2/3 complex (Schrodinger, 2010). Potential Van-der-Waals clashes were detected by show bumps plugin.

# Data availability

This study includes no data deposited in external repositories.

The source data of this paper are collected in the following database record: biostudies:S-SCDT-10_1038-S44319-024-00201-x.

# Peer review information

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

## Acknowledgements

This project has received funding from the European Research Council (ERC) under the European Union's Horizon 2020 research and innovation programme (grant agreement No 810207 to Michael Way. Luyan Cao was supported by the European Union's Horizon 2020 Marie Sklodowka-Curie individual fellowship programme (H2020-MSCA-IF-101028239—MolecularArp) and the ERC Synergy grant (No 810207). MW is additionally supported by the Francis Crick Institute, which receives its core funding from Cancer Research UK (CC2096), the UK Medical Research Council (CC2096), and the Wellcome Trust (CC2096). We thank Carolyn Moores (Birkbeck, University of London), Snezhka Oliferenko & Jeremy Carlton (the Francis Crick Institute, London) and Guillaume Romet-Lemonne & Antoine Jegou (Institut Jacques Monod, Paris) for feedback on the manuscript. For the purpose of Open Access, the authors have applied a CC BY public copyright licence to any Author Accepted Manuscript version arising from this submission.

## Author contributions

**LuYan Cao**: Conceptualisation; Resources; Formal analysis; Funding acquisition; Investigation; Methodology; Writing—original draft; Writing—review and editing. **Shaina Huang**: Resources; Formal analysis; Investigation; Methodology; Writing—review and editing. **Angika Basant**: Resources; Formal analysis; Investigation; Methodology; Writing—review and editing. **Miroslav Mladenov**: Resources. **Michael Way**: Conceptualization; Supervision; Funding acquisition; Writing—original draft; Writing—review and editing.

Source data underlying figure panels in this paper may have individual authorship assigned. Where available, figure panel/source data authorship is listed in the following database record: biostudies:S-SCDT-10_1038-S44319-024-00201-x.

## Funding

## Disclosure and competing interests statement

The authors declare no competing interests.

# Expanded View Figures

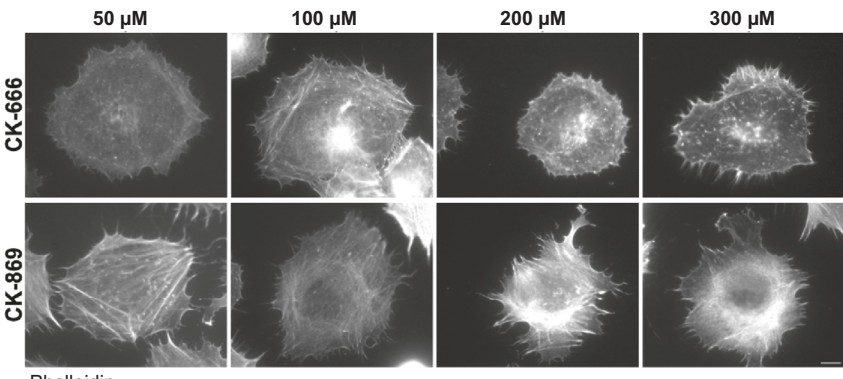

**Figure EV1.   CK-869, but not CK-666, fully inhibits Vaccinia-induced actin polymerisation.**

Immunofluorescence images of the actin cytoskeleton (visualised with phalloidin) of HeLa cells infected with Vaccinia virus at 9 h post-infection after 1-hour incubation with indicated concentrations of CK-666 and CK-869. The images correspond to the actin cytoskeleton of the cortactin images in Fig. 1B. Scale bar = 10 µm.

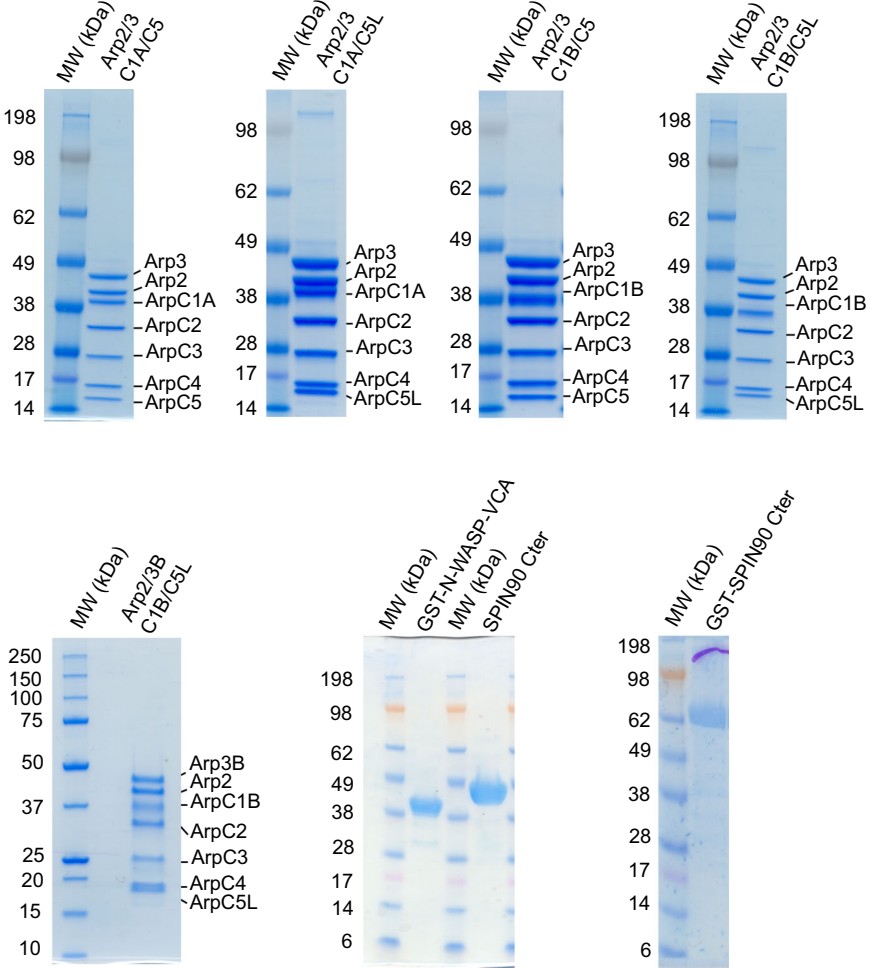

**Figure EV2.  Coomassie-stained protein gels of the recombinant proteins and Arp2/3 iso-complexes.**

Molecular weight markers and the protein names are indicated.

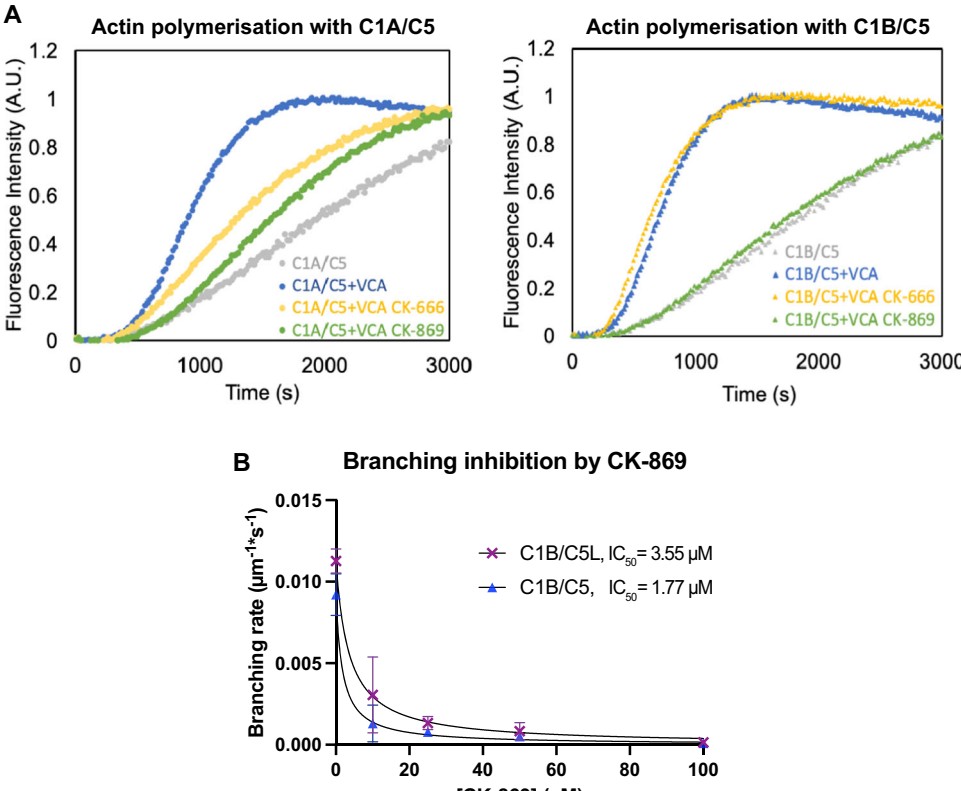

**Figure EV3. Inhibition of actin branches by CK-666 and CK-869.**

(A) Representative plots showing the polymerisation of pyrene actin (Fluorescence Intensity) when Arp2/3 complexes containing ArpC1A/C5 (left) or ArpC1B/C5 (right) are activated by GST-N-WASP in the absence (blue) or presence of 100 μM CK-666 (yellow) or CK-869 (green). (B) The branching rate of the Arp2/3 complex containing ArpC1B/C5 and ArpC1B/C5L (same as shown in Fig. 2C) were measured at the indicated CK-869 concentration. The data were fitted with equation Y = Bottom + (Top-Bottom)/(1 + (X/IC$_{50}$)) to calculate the half-maximal inhibitory concentration (IC$_{50}$) of the CK-869. The points and error bars represent the mean and the standard deviation of at least two independent measurements.

Gated on live CD11b⁺F4/80⁺ cells:

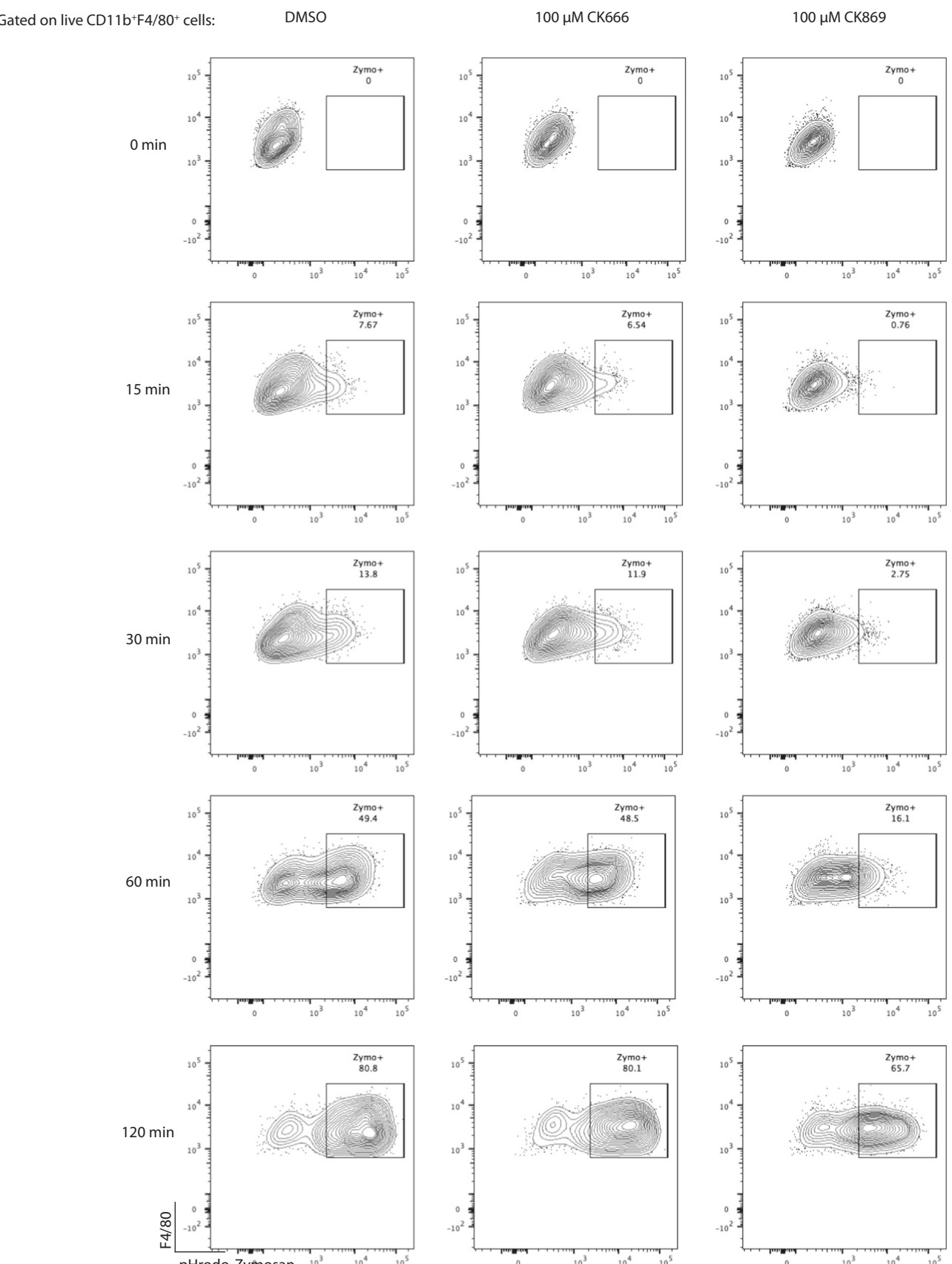

**Figure EV4.** Representative FACS plots of murine bone marrow-derived macrophage phagocytosis after treatment with DMSO (control) or 100 μM CK-666 or CK-869.

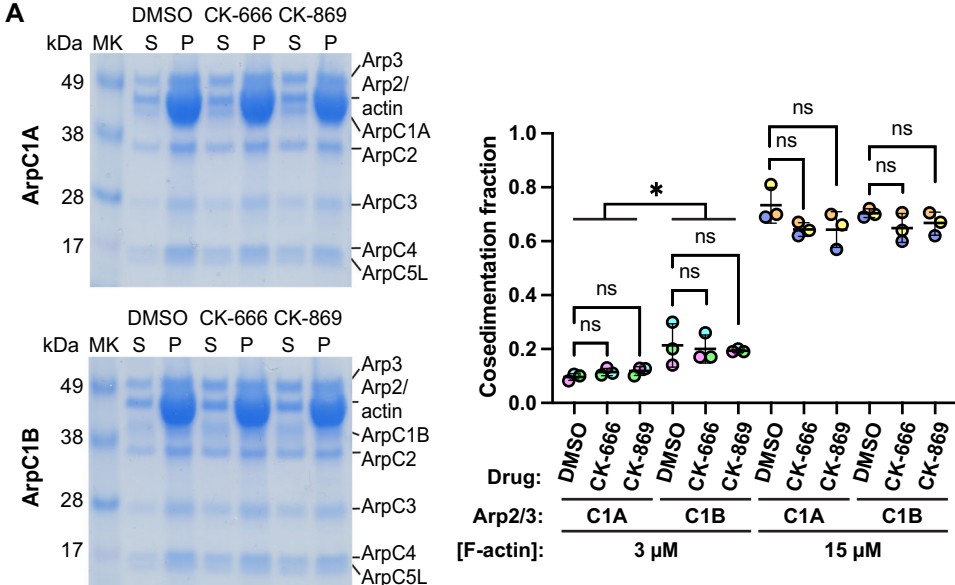

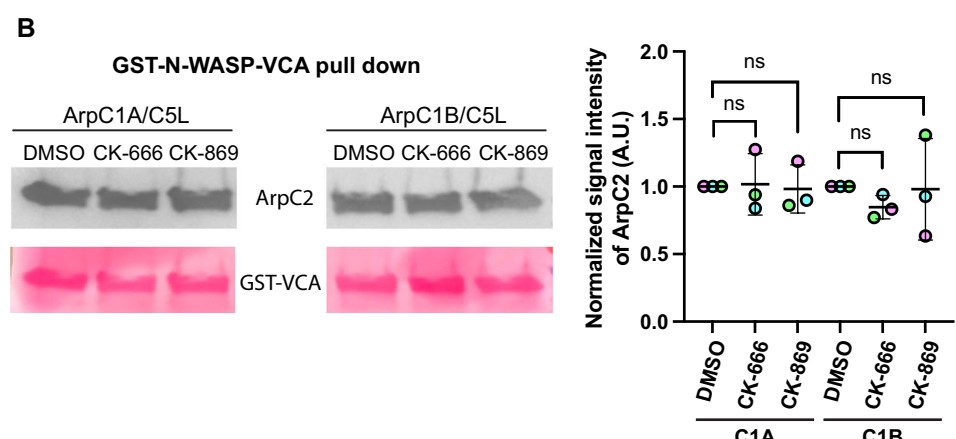

**Figure EV5. Analysis of Arp2/3 complex binding to actin filaments or VCA.**

(A) The left panel shows representative Coomassie-stained protein gels of the co-sedimentation of Arp2/3 complexes containing ArpC5L together with ArpC1A (top) or ArpC1B (bottom) with 15 µM actin (P pellet) in the absence (DMSO) or presence of 100 µM CK-666 or CK-869. The supernatant (S) contains unbound Arp2/3 complexes and non-pelleted G-actin. MK represents the molecular weight markers. The right graph shows the quantification of the mean co-sedimentation fraction of the indicated Arp2/3 complexes in the absence (DMSO) or presence of CK-666 or CK-869 from three independent experiments at 3 and 15 µM actin, with the error bars indicating the standard deviation. Two-tailed paired *t*-test was used to analyse the statistical significance. *$p$ value < 0.05. ns not significant. (B) Immunoblots (left panels) using an ArpC2 antibody demonstrate that 100 µM CK-666 or CK-869 does not impact the interaction of Arp2/3 complexes ArpC5L together with ArpC1A (top) or ArpC1B (bottom) with GST-N-WASP-VCA (Ponceau - red). The right graph shows the quantification of the mean pull-down fraction of the indicated Arp2/3 complexes in the absence (DMSO) or presence of CK-666 or CK-869 from three independent experiments, with the error bars indicating the standard deviation. Two-tailed paired *t*-test was used to analyse the statistical significance. ns not significant.

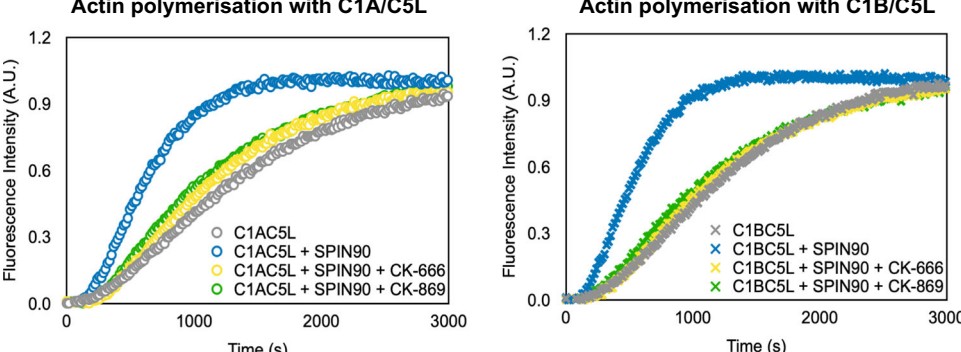

**Figure EV6.  Inhibition of SPIN90-Arp2/3 nucleated linear actin filaments.**

Representative plots showing the polymerisation of pyrene actin (Fluorescence Intensity) when Arp2/3 complexes containing ArpC1A/C5L (left) or ArpC1B/C5L (right) are activated by SPIN90-Cter in the absence (blue) or presence of 100 µM CK-666 (yellow) or CK-869 (green).

