## [Peer Review File · EMBO Reports]

CK-666 and CK-869 differentially inhibit Arp2/3 iso-complexes

LuYan Cao, Shaina Huang, Angika Basant, Miroslav Mladenov, and Michael Way

Corresponding author(s): Michael Way (Michael.Way@crick.ac.uk), LuYan Cao (luyan.cao@crick.ac.uk)

Review Timeline:

Transfer Date:	22nd Mar 24
Editorial Decision:	3rd May 24
Revision Received:	30th May 24
Accepted:	18th Jun 24

Editor: Deniz Senyilmaz Tiebe

**Transaction Report: This manuscript was transferred to
EMBO reports following peer review at Review Commons.**

**Review
COMMONS**

Review #1

1. Evidence, reproducibility and clarity:

Evidence, reproducibility and clarity (Required)

Summary:

This work focuses on two small molecule inhibitors of the Arp2/3 complex, CK-666 and CK-869. Previous studies have shown that although the Arp2/3 complex is well conserved in eukaryotes, the inhibitory effect of these molecules is highly species dependent. However, it has been unclear whether these drugs act equally well on Arp2/3 iso-complexes (complexes composed of subunit isoforms from the same species).

This paper fills that gap. Using human Arp2/3 iso-complexes, it shows that the inhibitory effect of these two drugs depends on the subunit composition of the complex. In addition, this work shows that these drugs do not systematically and equally inhibit the ability of these Arp2/3 complexes to nucleate linear or branched filaments.

Major comments:

1. Regarding the first part on vaccinia-induced actin polymerization

The first paragraph of the Results section is difficult to follow for those who have not read the previous papers from this lab. I would recommend changing the text so that any reader can understand from the start the experimental system and the goal of the experiment.

The data analysis of Figure 1C is not satisfactory. It is not very informative to statistically compare the effect of the two drugs at similar concentration. However, it is necessary to perform statistical tests to compare the different conditions with drug with the control condition (DMSO). By eye, I see a difference between DMSO and CK-666, so it is difficult to understand why the authors claim that CK-666 has no effect on actin polymerization.

Images with CK-869 have a lower overall cortactin signal, which could indicate that immunolabeling was not very effective in this condition. This could affect the analysis of the data in Figure 1C.

The authors mention that the exact levels of the 8 different Arp2/3 iso-complexes are not known in these HeLa cells, but it should be fairly easy (e.g. mass spectrometry) to quantify the expression level of ArpC1, ArpC5 and Arp3 in these cells and verify that it is consistent with the rest of the story.

This information about the expression level of ArpC1, ArpC5 and Arp3 in HeLa cells is also very important because a large community of researchers use CK-666 and HeLa cells. There are actually quite few papers that draw conclusions from the use of CK-666 in HeLa cells, and the authors should discuss the limitations of these studies much more clearly.

2. The pyrene assays are disappointing because they are performed with only one concentration of CK-666 and CK-869. This is especially true for the VCA data, where the effect of the drugs is not always "on"/"off" as naively presented in the text, but highly concentration dependent. The authors should definitely provide several drug concentrations for each condition, up to saturation levels, to provide a clear quantification of the drug concentrations needed to reach half inhibition.

3. Similarly, the pull-down experiments performed at a single protein concentration are inconclusive. They cannot tell us whether the affinity of the Arp2/3 isoforms for these targets is altered in the presence of the small molecule inhibitors because we do not know the degree of saturation of the ligands. Given that some of the reported differences in inhibition of filament nucleation are modest, it is not possible at this stage to link these different data.

2. Significance:

Significance (Required)

The subunit composition of the Arp2/3 complex is cell-type dependent, so these data will be important for the many cell biologists using these molecules. In particular, it calls for caution in the use of these drugs and in the interpretation of the data.

The writing is very clear, but the manuscript seems quite rushed. Many experiments need to be analyzed in much more detail to clarify the conclusions.

3. How much time do you estimate the authors will need to complete the suggested revisions:

Estimated time to Complete Revisions (Required)

(Decision Recommendation)

Between 3 and 6 months

Yes

Review #2

1. Evidence, reproducibility and clarity:

Evidence, reproducibility and clarity (Required)

The manuscript 'CK-666 and CK-869 differentially inhibit Arp2/3 iso-complexes' addresses how commonly used Arp2/3 complex inhibitors differentially inhibit Arp2/3 complex activity based on the subunit isoforms making up the Arp2/3 complex. This work directly tests how each inhibitor affects different iso-complexes, which may affect different cell types based on the predominant iso-complex present in the cell. The manuscript is well written, with experiments both in cell culture and with purified proteins in reconstitution and biochemical assays to establish that these small molecule inhibitors have different effects based on the iso-complex of Arp2/3 present. There are several points in the manuscript that if addressed would improve and support the conclusions presented.

In Figure 1B, looking at the images of the CK-666 treated versus the DMSO, it looks like the actin structures in the DMSO-treated cells are potentially larger than those in the CK666 cells, but because only an inset of drug-treated is shown, and an inset of the DMSO-treated is not shown it is hard to compare. Are the size of the virus-associated structures affected in the CK-666 treated cells versus the DMSO-treated cells? This might indicate that CK-666 has some effect on actin polymerization, even

if it is not as drastic as the CK-869.

In Figure 2 comparing the pyrene curves in figure 1A, it appears that CK-869 has a different effect on C1B/C5+VCA versus C1B/C5L+VCA (green curves as compared to no activation control, grey curves), but this is not commented on. Addressing the differing effects would strengthen the authors conclusions- namely, that CK-869 inhibits both iso-complexes better than CK-666, but there may be some differences on each isoform. It is unclear if the differences in the branching rate (Figure 2B) is also reflective of this. The authors should address these results.

For Figure 4, it is somewhat unexpected that inhibition of the Arp2/3 complex increases macrophage motility as compared to control, unless the reader is familiar with the 2017 Rotty et al paper. The manuscript may benefit from a sentence or two explaining this result in light of the findings of the 2017 Rotty paper beyond simply mentioning that the increase in motility is dependent on myosin II.

The Spin90 data looks good, clear, and consistent.

In Figure 7, given that pyrene was used in all the previous assessments of drug treatment on arp2/3 isoforms, it seems appropriate for these assays to be performed for Arp3B/C1B/C5L in comparison with Arp3/C1B/C5L and between the different drug treatments. Likewise, this should be done for the Spin90 also. It is difficult to compare between the figures for Arp3b vs. Arp3C (Figures 2 and 3 vs. Figure 7), although this may require a repetition of data presented.

****Minor issues:****

It would be helpful if the labels for what is labeled in the micrograph were on the images (Figure 1B, Figure 3B, Figure 7A).

In Figure 1-B, the 200uM CK-869 cell image looks less representative of the data in Figure 1C than other cells in the figure. Perhaps there is higher background in this micrograph, but it might be clearer if a cell with similar background actin signal to the other CK-869 was used.

Figure 4: Where is the mean thickness of the cell measured? In figure 4D, it would be helpful if the error bars could be in the color of the line, as it is hard to distinguish the range of the data for each condition because the error bars are overlapping and the same color for all.

Figure 5: In Figure 5A, the labeling of the gels (KDa, S and P) do not line up

correctly. The legend for the quantification should indicate what bands were quantified- all the arp2/3 bands or just the isoforms? It is unclear what is being quantified in the graph in C. The pull-down results in C should be quantified via quantitative western blot if possible.

The statement in line 170- 'indicate' seems a bit strong based on the results presented. 'Suggests' might work better here.

2. Significance:

Significance (Required)

It is of general interest to members of the actin field as well as cell-biologists who routinely use either CK-666 or CK-869 to inhibit Arp2/3 complex activity in cells, and specifically in mammalian cells.

3. How much time do you estimate the authors will need to complete the suggested revisions:

Estimated time to Complete Revisions (Required)

(Decision Recommendation)

Less than 1 month

Yes

Review #3

1. Evidence, reproducibility and clarity:

Evidence, reproducibility and clarity (Required)

****Summary:****

Cao et al combine in vitro and cellular work to show that neither of the two distinct and frequently used Arp2/3 inhibitors is truly pan-selective, at least when considering distinct classes of activators. Using in vitro assays, they show that CK-666 cannot inhibit ARPC1B iso-complexes when activated by class I nucleation promoting factors. Similarly, Arp2/3 complexes containing Arp3B are refractory to inhibition by CK-869. The latter is likely the result of substitutions at the inhibitor-binding site. They go on to show that these differences correlate with differential effects of CK-666 and -869 on Vaccinia tail formation and macrophage cell shape and motility at the cellular level.

****Major comments:****

Figure1: The authors state that "...even at 300 μ M, the number of virus-induced actin polymerisation events were not diminished (Figure 1B, C)..." The figure shows that CK-666 does indeed not fully abolish cortactin colocalization. However, there seems to still be a significant effect that is not tested for. Statistical tests were only used to compare the two inhibitors at the same concentration. I suggest also testing for significant differences to the DMSO control and reporting p-values, because CK-666 seems to still have an effect. Along the same vein, it seems that valuating the fraction of virus with cortactin co-localization as the only metric for branched actin nucleation downplays the effects of CK-666. Can the authors consider additional other metrics such as the amount of polymerized actin in individual tails or the tail length, which were extensively used in previous publications?

Figure2/3: The authors claim that "...the ArpC5/ArpC5L isoforms are not differentially impacted by either CK-666 or CK-869..." I am not convinced that this conclusion can be drawn based on the data. Figure 2 shows that the inhibitory effect of CK-869 seems to be less pronounced for C5L-containing complexes (about 10-fold reduced branching rate) compare to C5-containing ones (about 100-fold reduction). This is in line with the pyrene assays, in which C5L-containing complexes (in contrast to C-5) appear to retain at least some activity. Differences should be quantified relative to the corresponding controls and then statistically tested for using appropriate

tests.

Figure 4: Cell metrics such as aspect ratio (A), thickness (A) and speed (C) are expressed as means from five independent experiments. It is not clear how many individual cells were scored per experiment per condition. Similarly, it is unclear at which time (or time window) after inhibitor addition these parameters were scored. Claiming that the authors "...observed that the morphology of macrophages treated with CK-869 changed significantly, with cells rounding up to become less spread..." is a slight over-interpretation, because these metrics have not been quantified in a time-resolved manner but only as a snapshot of the population mean.

****Minor:****

Figure 2/3: In my opinion, separating the in vitro data for ARPC1A/B containing sub-complexes and starting with B does not work particularly well for the flow paper. The results for the C1A containing Arp2/3 complexes (Figure 3) essentially confirm that both inhibitors work at least on some, but not all iso-complexes, as they should. These experiments are -in a way- necessary controls for those shown in the previous figure (Figure 2). I would suggest merging the two figures and starting with the less surprising findings with 1A before showing differential inhibition on 1B.

Figure 2/3: Inhibitor concentrations should be stated in the figure and not only in the legend.

Figure 4B: The macrophages shown for the CK-869 treatment appear less spread and more round already at $t=0$ (before inhibitor application), although this is hard to tell for the low contrast PC images. I would recommend showing either images of comparable contrast and cell spread area at $t=0$ or change to live cell marker and fluorescence imaging.

Figure 5A: Left and right sub-panels should contain clear labels on top indicating which iso-complexes are being examined (1A left, 1B right). Please also clearly state the total concentrations of actin and Arp2/3 complex used in the figure legend. The low fraction (<20%) of Arp2/3 complex co-sedimenting with actin filaments is rather surprising considering the high concentrations used here. 7.5 μ M actin should be well above the KD for this interaction (compare to data of the Nolen lab such as Hetrick et al 2013). Please comment.

****Referee Cross-commenting****

The reviews appear to be quite consistent, highlighting several critical issues mentioned by multiple referees. While all referees appreciate the topic/focus of the

manuscript, they criticize its preliminary nature.

I anticipate that this would lead to a "major revision" decision at a traditional journal. The numerous constructive comments should enable the authors to significantly enhance the paper if taken seriously.

2. Significance:

Significance (Required)

Small molecule inhibitors such as CK-666 and -869 have been (and still are) widely utilized in the cytoskeleton community as straightforward tools to suppress Arp2/3 activity. However, the results presented here emphasize the need for caution in drawing simplistic conclusions. Hence, future interpretations must adopt a more nuanced perspective. The manuscript therefore makes an important, timely contribution and will be of great interest to a large community.

In terms of its potential impact, it reminds me of the recent cautionary tale showing that small molecule formin inhibitors have significant off-target effects (Nishimura et al JCS 2021). However, it is crucial to note that isoform specificity differs from off-target effects, and this doesn't necessarily implicate CK-666 and -869 as inadequate inhibitors.

While the manuscript is technically sound, with carefully conducted experiments, the presentation and writing seem rushed at times, warranting improvement before publication. The points highlighted above are intended to enhance the overall quality.

Conceptually, one central weakness is that the reason for the differential inhibition remains ultimately unclear at least in some cases. Specifically, why ARPC1B complexes are refractory to CK-666 inhibition when activated by class I NPFs is not known. Similarly, why activation by different inputs (SPIN90 vs NPFs) is differentially sensitive to different inhibitors remains unclear. Addressing these gaps through additional experiments would strengthen the study. A more insightful discussion, drawing on existing structural and biochemical data, even if speculative, would also be helpful in this regard.

Own expertise: cytoskeleton, actin, biochemistry, in vitro reconstitution, fluorescence microscopy, structural biology

Signed: Peter Bieling, MPI Dortmund

3. How much time do you estimate the authors will need to complete the suggested revisions:

Estimated time to Complete Revisions (Required)

(Decision Recommendation)

Between 1 and 3 months

Yes

Full Revision

Manuscript number: RC-2023-02277

Corresponding author(s): LuYan Cao and Michael Way

1. General Statements [optional]

The inhibitors, CK-666 and CK-869, are widely used to probe the function of actin nucleation by the Arp2/3 complex *in vitro* and in cells. However, in mammals, the Arp2/3 complex consists of 8 iso-complexes, which have different properties. Using recombinant Arp2/3 with defined composition we have demonstrated that the inhibitory effect of CK-666 and CK-869 depends on the Arp2/3 iso-complexes and whether they are activated by VCA or SPIN90. Our findings have important implications for the interpretation of results using CK-666 and CK-869, given that the relative expression levels of ArpC1 and Arp3 isoforms in cells and tissues remains largely unknown.

We are pleased that all three reviewers appreciated the importance of our observations, had similar questions/concerns and were positive. Moreover, their additional questions, which we have now fully addressed have significantly improved our study and helped better clarify our conclusions.

This section is mandatory. Please insert a point-by-point reply describing the revisions that were already carried out and included in the transferred manuscript.

Reviewer #1 (Evidence, reproducibility and clarity (Required)):

Summary:

This work focuses on two small molecule inhibitors of the Arp2/3 complex, CK-666 and CK-869. Previous studies have shown that although the Arp2/3 complex is well conserved in eukaryotes, the inhibitory effect of these molecules is highly species dependent. However, it has been unclear whether these drugs act equally well on Arp2/3 iso-complexes (complexes composed of subunit isoforms from the same species). This paper fills that gap. Using human Arp2/3 iso-complexes, it shows that the inhibitory effect of these two drugs depends on the subunit composition of the complex. In addition, this work shows that these drugs do not systematically and equally inhibit the ability of these Arp2/3 complexes to nucleate linear or branched filaments.

We thank the reviewer for their positive comments.

Major comments:

1/ Regarding the first part on vaccinia-induced actin polymerization

The first paragraph of the Results section is difficult to follow for those who have not read the previous papers from this lab. I would recommend changing the text so that any reader can understand from the start the experimental system and the goal of the experiment.

As requested, we have expanded the first section to better allow the reader to understand vaccinia actin-based motility as a model system to understand Arp2/3 iso-complex function.

The data analysis of Figure 1C is not satisfactory. It is not very informative to statistically compare the effect of the two drugs at similar concentration. However, it is necessary to perform statistical tests to compare the different conditions with drug with the control condition (DMSO). By eye, I see a difference between DMSO and CK-666, so it is difficult to understand why the authors claim that CK-666 has no effect on actin polymerization.

We are claiming that CK-869 but not CK-666 fully inhibits the ability of Vaccinia virus to stimulate Arp2/3 dependent actin polymerisation. We agree that CK-666 partially inhibits Vaccinia induced actin polymerization and have changed the text accordingly to reflect this. In contrast to CK-869 this level of inhibition does change from that seen with 50 μ M when CK-666 is increased up to 300 μ M. We believe this partial ~30% inhibition reflects the impact of CK-666 inhibiting ArpC1A containing Arp2/3 from generating actin filaments. These inhibited ArpC1A containing Arp2/3 complexes are able to bind the VCA domain of N-WASP (see figure 4) which will block its interaction with ArpC1B containing complexes. We now have provided the requested statistical analysis between drug and DMSO and also retained our original statistical analysis between the drugs.

Images with CK-869 have a lower overall cortactin signal, which could indicate that immunolabeling was not very effective in this condition. This could affect the analysis of the data in Figure 1C.

In the figure we used cortactin as a marker for branched actin filaments to assess the impact of CK-666 and CK-869 on the ability of individual vaccinia viruses to induce actin polymerization rather than the extent of actin assembly. In general, CK-869 does not impact on cortactin signal, however, the differences the reviewer is referring to are probably due to cell-to-cell variability. Moreover, we have now provided the corresponding image of the actin visualized with phalloidin as supplementary figure 1. In these images the same virus induced actin structures are visible.

The authors mention that the exact levels of the 8 different Arp2/3 iso-complexes are not known in these HeLa cells, but it should be fairly easy (e.g. mass spectrometry) to quantify the expression level of ArpC1, ArpC5 and Arp3 in these cells and verify that it is consistent with the rest of the story.

This information about the expression level of ArpC1, ArpC5 and Arp3 in HeLa cells is also very important because a large community of researchers use CK-666 and HeLa cells. There are actually quite few papers that draw conclusions from the use of CK-666 in HeLa cells, and the authors should discuss the limitations of these studies much more clearly.

In Abella et al. NCB 2016 we quantified the amounts of ARPC1 and ARPC5 isoforms in our HeLa cell. ArpC1A is 0.3 ± 0.02 ng/ μ g cells; ArpC1B is 0.7 ± 0.05 ng/ μ g cells; ArpC5 is 0.46 ± 0.03 ng/ μ g cells; ArpC5L is 0.27 ± 0.03 ng/ μ g cells. Thus, ArpC1B is approximately twice that of ARPC1A which fits with the ~30 % level of inhibition we see with CK-666 in figure 1C. Unfortunately, we do not have a specific antibody against Arp3B, so have not been able to use the same approach to quantify the level of this isoform. However, Arp3B is 18.5-fold less abundant than Arp3 in HeLa cells according to Hein et al., 2015 (PMID: 26496610 DOI: 10.1016/j.cell.2015.09.053). In an early study (Kulak et al., 2014 PMID: 24487582 DOI: 10.1038/nmeth.2834, the same group reported that Arp3 was 61.5 X more abundant than Arp3B in HeLa cells. These two papers illustrate the difficulty in using mass spec to determine absolute protein concentrations, which is why we prefer quantitative western blotting as done in Abella et al., 2016.

2/ The pyrene assays are disappointing because they are performed with only one concentration of CK-666 and CK-869. This is especially true for the VCA data, where the effect of the drugs is not always "on"/"off" as naively presented in the text, but highly concentration dependent. The authors should definitely provide several drug concentrations for each condition, up to saturation levels, to provide a clear quantification of the drug concentrations needed to reach half inhibition.

Following the reviewer's advice, we now have performed the pyrene and TIRF assays in the presence of a range of drug concentrations (see individual figures). These new data have allowed us to calculate the half-maximal inhibitory concentration values (IC_{50}) which strengthen our previous conclusions. CK-666 can prevent ArpC1A ($IC_{50} = 20 \mu M$) but not ArpC1B (IC_{50} undetectable) from generating branches. Meanwhile, CK-869 can inhibit both ArpC1 isoforms efficiently with $IC_{50} < 10 \mu M$. The inhibition of CK-869 against Arp3B containing complex is not measurable.

3/ Similarly, the pull-down experiments performed at a single protein concentration are inconclusive. They cannot tell us whether the affinity of the Arp2/3 isoforms for these targets is altered in the presence of the small molecule inhibitors because we do not know the degree of saturation of the ligands. Given that some of the reported differences in inhibition of filament nucleation are modest, it is not possible at this stage to link these different data.

Following the reviewer's advice, we repeated the pull down Arp2/3 at a higher F-actin concentration. In the initial submission we said we used $7.5 \mu M$ F-actin, however, we discovered a miscalculation, so it was actually $3 \mu M$, which would explain the lower levels of Arp2/3 co-

Full Revision

pelleting. In Hetrick et al 2013 (PMID: 23623350 DOI: 10.1016/j.chembiol.2013.03.019), the binding of Arp2/3 to F actin reaches a plateau at 15 μ M F-actin. We therefore used 15 μ M F-actin for the additional pull down experiments (Figure 4). The new results with 15 μ M F-actin agree with our previous observations at 3 μ M F-actin concentration.

We do not feel it is necessary to repeat the pull down of Arp2/3 by GST-VCA at different concentrations. This is because Arp2/3 binds VCA with high affinity (0.9 μ M) Marchand et al. NCB 2001 (PMID: 11146629 DOI: 10.1038/35050590). Thus in our initial experimental conditions (5 μ M VCA), the binding is already saturated. In addition, we did not see a difference in binding between Arp2/3 iso-complexes.

Reviewer #1 (Significance (Required)):

The subunit composition of the Arp2/3 complex is cell-type dependent, so these data will be important for the many cell biologists using these molecules. In particular, it calls for caution in the use of these drugs and in the interpretation of the data.

The writing is very clear, but the manuscript seems quite rushed. Many experiments need to be analyzed in much more detail to clarify the conclusions.

We thank the reviewer for their positive comments and suggestions to improve our study.

Reviewer #2 (Evidence, reproducibility and clarity (Required)):

The manuscript 'CK-666 and CK-869 differentially inhibit Arp2/3 iso-complexes' addresses how commonly used Arp2/3 complex inhibitors differentially inhibit Arp2/3 complex activity based on the subunit isoforms making up the Arp2/3 complex. This work directly tests how each inhibitor affects different iso-complexes, which may affect different cell types based on the predominant iso-complex present in the cell. The manuscript is well written, with experiments both in cell culture and with purified proteins in reconstitution and biochemical assays to establish that these small molecule inhibitors have different effects based on the iso-complex of Arp2/3 present. There are several points in the manuscript that if addressed would improve and support the conclusions presented.

We thank the reviewer for their comments and suggestions.

In Figure 1B, looking at the images of the CK-666 treated verses the DMSO, it looks like the actin structures in the DMSO-treated cells are potentially larger than those in the CK666 cells, but because only an inset of drug-treated is shown, and an inset of the DMSO-treated is not shown it is hard to compare. Are the size of the virus-associated structures affected in the CK-666 treated cells versus the DMSO-treated cells? This might indicate that CK-666 has some effect on actin polymerization, even if it is not as drastic as the CK-869.

Full Revision

The reviewer is right that actin tails are shorter in CK-666 treated cells. This is because CK-666 does partially inhibit actin polymerisation induced by the virus. In contrast to CK-869, this level of inhibition does change with increasing concentration of CK-666. We believe this partial inhibition reflects the impact of CK-666 inhibiting ArpC1A containing Arp2/3 from generating actin filaments. These inhibited complexes will bind the VCA domain of N-WASP (see figure 4) blocking its interaction with ArpC1B containing complexes.

In Figure 2 comparing the pyrene curves in figure 1A, it appears that CK-869 has a different effect on C1B/C5+VCA versus C1B/C5L+VCA (green curves as compared to no activation control, grey curves), but this is not commented on. Addressing the differing effects would strengthen the authors conclusions- namely, that CK-869 inhibits both iso-complexes better than CK-666, but there may be some differences on each isoform. It is unclear if the differences in the branching rate (Figure 2B) is also reflective of this. The authors should address these results.

This is a very good point. We have now performed more detailed analysis, measuring the branching rate of C1B/C5 and C1B/C5L complexes in TIRF assays in different concentrations of CK-869 (Supplementary figure 2B). By comparing the half-maximal inhibitory concentration values (IC_{50}) of CK-869 on the two different complexes, we found CK-869 inhibits C1B/C5 slightly better than C1B/C5L (1.8 μ M as compared to 3.6 μ M) as the reviewer suggested.

For Figure 4, it is somewhat unexpected that inhibition of the Arp2/3 complex increases macrophage motility as compared to control, unless the reader is familiar with the 2017 Rotty et al paper. The manuscript may benefit from a sentence or two explaining this result in light of the findings of the 2017 Rotty paper beyond simply mentioning that the increase in motility is dependent on myosin II.

As requested, we have provided more information.

The Spin90 data looks good, clear, and consistent.

We thank reviewer for the positive comments.

In Figure 7, given that pyrene was used in all the previous assessments of drug treatment on arp2/3 isoforms, it seems appropriate for these assays to be performed for Arp3B/C1B/C5L in comparison with Arp3/C1B/C5L and between the different drug treatments. Likewise, this should be done for the Spin90 also. It is difficult to compare between the figures for Arp3b vs. Arp3C (Figures 2 and 3 vs. Figure 7), although this may require a repetition of data presented.

We have now provided quantification of the maximum actin polymerization rate induced by Arp3B/C1B/C5L complexes obtained in pyrene assembly assays over a range of drug concentrations (requested by reviewer 1) (Figure 6C). These new data confirm that Arp3B is not

Full Revision

inhibited by CK-869. We did not feel it was necessary to perform a side-by-side comparison with Arp3/C1B/C5L complexes but have provided quantification of the branching rate of Arp3/C1B/C5L complexes over a range of drug concentrations using TIRF assays (see Figure 2D).

Minor issues:

It would be helpful if the labels for what is labeled in the micrograph were on the images (Figure 1B, Figure 3B, Figure 7A).

We have provided the requested labels.

In Figure 1-B, the 200uM CK-869 cell image looks less representative of the data in Figure 1C than other cells in the figure. Perhaps there is higher background in this micrograph, but it might be clearer if a cell with similar background actin signal to the other CK-869 was used.

As we responded to reviewer 1: In the figure we used cortactin as a marker for branched actin filaments to assess the impact of CK-666 and CK-869 on the ability of individual vaccinia viruses to induce actin polymerization rather than the extent of actin assembly. In general, CK-869 does not impact on cortactin signal, however, the differences the reviewer is referring to are probably due to cell-to-cell variability. Moreover, we have now provided the corresponding image of the actin channel visualised with phalloidin as supplementary figure 1. In these images the same virus induced actin structures are visible.

Figure 4: Where is the mean thickness of the cell measured? In figure 4D, it would be helpful if the error bars could be in the color of the line, as it is hard to distinguish the range of the data for each condition because the error bars are overlapping and the same color for all.

We used the Phasefocus Liveocyte to image and quantify the morphology and behaviour of live cells. The mean thickness of the cell is quantified from the whole cell area based on the method described in Marrison et al. Scientific reports 2013 (PMID: 23917865 DOI: 10.1038/srep02369). We have clarified this fact in the figure legend. We have also corrected the colour issue with the error bars.

Figure 5: In Figure 5A, the labeling of the gels (KDa, S and P) do not line up correctly. The legend for the quantification should indicate what bands were quantified- all the arp2/3 bands or just the isoforms? It is unclear what is being quantified in the graph in C. The pull-down results in C should be quantified via quantitative western blot if possible.

We have provided new F-actin pulldown gels and have made sure the labels are aligned. The level of Arp2/3 binding to F-actin was determined by quantifying the level of bound ArpC3. This subunit was chosen as it is well removed from the other bands on the gel. We have now also provided quantification of the VCA pulldowns assays as requested.

Full Revision

The statement in line 170- 'indicate' seems a bit strong based on the results presented. 'Suggests' might work better here.

We have changed the text as suggested by the reviewer.

Reviewer #2 (Significance (Required)):

It is of general interest to members of the actin field as well as cell-biologists who routinely use either CK-666 or CK-869 to inhibit Arp2/3 complex activity in cells, and specifically in mammalian cells.

We thank the reviewer for their positive comments and suggestions.

Reviewer #3 (Evidence, reproducibility and clarity (Required)):

Summary:

Cao et al combine in vitro and cellular work to show that neither of the two distinct and frequently used Arp2/3 inhibitors is truly pan-selective, at least when considering distinct classes of activators. Using in vitro assays, they show that CK-666 cannot inhibit ARPC1B iso-complexes when activated by class I nucleation promoting factors. Similarly, Arp2/3 complexes containing Arp3B are refractory to inhibition by CK-869. The latter is likely the result of substitutions at the inhibitor-binding site. They go on to show that these differences correlate with differential effects of CK-666 and -869 on Vaccinia tail formation and macrophage cell shape and motility at the cellular level.

Major comments:

Figure1: The authors state that "...even at 300 μ M, the number of virus-induced actin polymerisation events were not diminished (Figure 1B, C)..." The figure shows that CK-666 does indeed not fully abolish cortactin colocalization. However, there seems to still be a significant effect that is not tested for. Statistical tests were only used to compare the two inhibitors at the same concentration. I suggest also testing for significant differences to the DMSO control and reporting p-values, because CK-666 seems to still have an effect. Along the same vein, it seems that valuating the fraction of virus with cortactin co-localization as the only metric for branched actin nucleation downplays the effects of CK-666. Can the authors consider additional other metrics such as the amount of polymerized actin in individual tails or the tail length, which were extensively used in previous publications?

This point was also raised by the two other reviewers (see above). We have now provided the requested statistical tests. In our study we used Vaccinia as a model to examine whether there were differences between the impact of CK-666 and CK-869 on Arp2/3 dependent actin

polymerization in cells. This is clearly the case, so we focused on in vitro assays where we can do experiments with defined Arp2/3 iso-complexes to better understand what was going on. Given the complexity of cellular systems, we feel that additional analysis of the changes to the actin tails will not provide additional molecular insights, especially as the factors that determine actin tail lengths are still not fully understood.

Figure 2/3: The authors claim that "...the ArpC5/ArpC5L isoforms are not differentially impacted by either CK-666 or CK-869..." I am not convinced that this conclusion can be drawn based on the data. Figure 2 shows that the inhibitory effect of CK-869 seems to be less pronounced for C5L-containing complexes (about 10-fold reduced branching rate) compare to C5-containing ones (about 100-fold reduction). This is in line with the pyrene assays, in which C5L-containing complexes (in contrast to C-5) appear to retain at least some activity. Differences should be quantified relative to the corresponding controls and then statistically tested for using appropriate tests.

This point was also raised by reviewer 2. We have now performed more detailed analysis, measuring the branching rate of C1B/C5 and C1B/C5L complexes in TIRF assays in different concentrations of CK-869 (Supplementary figure 2B). By comparing the half-maximal inhibitory concentration values (IC50) of CK-869 on the two different complexes, we found CK-869 inhibits C1B/C5 slightly better than C1B/C5L (1.8 μM as compared to 3.6 μM) as the reviewer suggested.

Figure 4: Cell metrics such as aspect ratio (A), thickness (A) and speed (C) are expressed as means from five independent experiments. It is not clear how many individual cells were scored per experiment per condition. Similarly, it is unclear at which time (or time window) after inhibitor addition these parameters were scored. Claiming that the authors "...observed that the morphology of macrophages treated with CK-869 changed significantly, with cells rounding up to become less spread..." is a slight over-interpretation, because these metrics have not been quantified in a time-resolved manner but only as a snapshot of the population mean.

We have now provided the number of cells analysed in the individual experiments in figure 4. All measurements were taken after incubating cells for 1 hour with the Arp2/3 inhibitors, which is commonly used for cell-based experiments. We have now also provided a movie (Phase and GFP-LifeAct) covering 5 hours immediately after treating cells with DMSO or 100 μM CK-666 / CK-869 for 1 hour showing that the cell morphology does not change during the imaging period.

Minor:

Figure 2/3: In my opinion, separating the in vitro data for ARPC1A/B containing sub-complexes and starting with B does not work particularly well for the flow paper. The results for the C1A containing Arp2/3 complexes (Figure 3) essentially confirm that both inhibitors work at least on some, but not all iso-complexes, as they should. These experiments are -in a way- necessary controls for those shown in the previous figure (Figure 2). I would suggest merging the two

Full Revision

figures and starting with the less surprising findings with 1A before showing differential inhibition on 1B.

We have now merge both figure 2 and 3, with some of the original pyrene panels being moved into supplemental figure 2. The new figure 2 also contains quantification of the branching rate in different drug concentrations.

Figure 2/3: Inhibitor concentrations should be stated in the figure and not only in the legend.

This information has been added to the figure.

Figure 4B: The macrophages shown for the CK-869 treatment appear less spread and more round already at $t=0$ (before inhibitor application), although this is hard to tell for the low contrast PC images. I would recommend showing either images of comparable contrast and cell spread area at $t=0$ or change to live cell marker and fluorescence imaging.

$T=0$ is at the point of live cell imaging of cells which have already been treated with the Arp2/3 inhibitors for 1 hour. Consequently, the cells will never appear spread in the CK-869 treated sample. We have provided the fluorescent channel (GFP-LifeAct) in the movie.

Figure 5A: Left and right sub-panels should contain clear labels on top indicating which iso-complexes are being examined (1A left, 1B right). Please also clearly state the total concentrations of actin and Arp2/3 complex used in the figure legend. The low fraction (<20%) of Arp2/3 complex co-sedimenting with actin filaments is rather surprising considering the high concentrations used here. 7.5 μ M actin should be well above the KD for this interaction (compare to data of the Nolen lab such as Hetrick et al 2013). Please comment.

Thank reviewer for this suggestion and the requested information is provided. The reviewer is totally correct as we found that there was calculation error and the final actin concentration was actually 3 μ M and not 7.5 μ M as we originally thought.

In Hetrick et al 2013 (PMID: 23623350 DOI: 10.1016/j.chembiol.2013.03.019), the binding of Arp2/3 to F actin reaches a plateau at 15 μ M F-actin. We therefore used 15 μ M F-actin for the additional pull down experiments requested by reviewer 1 (Figure 4). The new results with 15 μ M F-actin agree with our previous observations at 3 μ M F-actin concentration.

****Referee Cross-commenting****

The reviews appear to be quite consistent, highlighting several critical issues mentioned by multiple referees. While all referees appreciate the topic/focus of the manuscript, they criticize its preliminary nature.

I anticipate that this would lead to a "major revision" decision at a traditional journal. The

Full Revision

numerous constructive comments should enable the authors to significantly enhance the paper if taken seriously.

All three reviewers had similar issues, which we believe we have now fully addressed.

Reviewer #3 (Significance (Required)):

Significance:

Small molecule inhibitors such as CK-666 and -869 have been (and still are) widely utilized in the cytoskeleton community as straightforward tools to suppress Arp2/3 activity. However, the results presented here emphasize the need for caution in drawing simplistic conclusions. Hence, future interpretations must adopt a more nuanced perspective. The manuscript therefore makes an important, timely contribution and will be of great interest to a large community.

We thank the reviewer for their positive assessment.

In terms of its potential impact, it reminds me of the recent cautionary tale showing that small molecule formin inhibitors have significant off-target effects (Nishimura et al JCS 2021). However, it is crucial to note that isoform specificity differs from off-target effects, and this doesn't necessarily implicate CK-666 and -869 as inadequate inhibitors.

We agree with the reviewer that these are still useful inhibitors.

While the manuscript is technically sound, with carefully conducted experiments, the presentation and writing seem rushed at times, warranting improvement before publication. The points highlighted above are intended to enhance the overall quality.

The reviewer's points and comments have definitely helped improve the study.

Conceptually, one central weakness is that the reason for the differential inhibition remains ultimately unclear at least in some cases. Specifically, why ARPC1B complexes are refractory to CK-666 inhibition when activated by class I NPFs is not known. Similarly, why activation by different inputs (SPIN90 vs NPFs) is differentially sensitive to different inhibitors remains unclear. Addressing these gaps through additional experiments would strengthen the study. A more insightful discussion, drawing on existing structural and biochemical data, even if speculative, would also be helpful in this regard.

We agree that we still lack a full molecular understanding for the differences but feel that getting to that point will require a substantial amount of work and new Xtal structures that are beyond the scope of the current work. However, we have updated our discussion drawing on existing data as requested by the reviewer.

Dear Dr. Cao,
Dear Michael,

Thank you for submitting your revised manuscript, which was previously peer reviewed at Review Commons. It has now been seen by all of the original referees.

As you can see, the referees find that the study is significantly improved during revision and recommend publication. However, I need you to address the points below before I can accept the manuscript.

- Please address the remaining minor concern of Referee #1.
- Please submit source data as requested by our Source Data Coordinator Dr. Hannah Sonntag.
- Your manuscript is better suited for our Scientific Report article type, which allows max 5 main figures as per our format requirements. Therefore, please convert one of the main figures into an EV Figure, and update the figure callouts accordingly (please see <https://www.embopress.org/page/journal/14693178/authorguide#researcharticleguide>).
- Please submit the manuscript text in word format.
- Please remove the Conclusions title and include section in the Results and discussion section.
- Please address the remaining minor concern of referee #1.
- Please provide 3-5 keywords for your study. These will be visible in the html version of the paper and on PubMed and will help increase the discoverability of your work.
- As per our guidelines, please add a 'Data Availability' section, where you state that no data were deposited in a public database.
- Please add a "Disclosure Statement and Competing Interests" section (<https://www.embopress.org/page/journal/14693178/authorguide#conflictsofinterest>).
- Please fill out and include an author checklist as listed in our online guidelines (<https://www.embopress.org/page/journal/14693178/authorguide>)
- Please re-submit all main figures and EV figures separately as individual production quality Figure files (<https://www.embopress.org/page/journal/14693178/authorguide#figureformat>).
- As per the movie, please rename it as "Movie EV1" and remove its legend from the manuscript file. Instead, please provide it in a readme.txt file as "Movie EV1: legend text" and zip up both the legend and the movie file to upload Movie. The callout in the manuscript should also be updated accordingly.
- The manuscript sections should be in the following order: Title page - Abstract & Keywords - Introduction - Results - Discussion - Methods - Data Availability - Acknowledgments - Disclosure Statement & Competing Interests - References - Figure Legends - Expanded View Figure Legends.
- Supplementary Figures, Legends and their callouts need to be renamed as Expanded View Figures (Figure EV1, etc.) and the callouts need to be updated in the manuscript text accordingly.
- Our production/data editors have asked you to clarify several points in the figure legends:
 - o Please note that a separate 'Data Information' section is required in the legends of figures 3a, c-d. (please see <https://www.embopress.org/page/journal/14693178/authorguide#figureformat> for an example)
 - o Please note that in figures 1c; 2b; 3d; there is a mismatch between the annotated p values in the figure legend and the annotated p values in the figure file that should be corrected.
- Papers published in EMBO Reports include a 'synopsis' and 'bullet points' to further enhance discoverability. Both are displayed on the html version of the paper and are freely accessible to all readers. The synopsis includes a short standfirst summarizing the study in 1 or 2 sentences (max 35 words) that summarize the paper and are provided by the authors and streamlined by the handling editor. I would therefore ask you to include your synopsis blurb and 3-5 bullet points listing the key experimental findings.
- In addition, please provide an image for the synopsis. This image should provide a rapid overview of the question addressed in the study but still needs to be kept fairly modest since the image size cannot exceed 550 (width) x 300-600 (height) pixels.

Thank you again for giving us to consider your manuscript for EMBO Reports, I look forward to your minor revision.

Kind regards,

Deniz

--

Deniz Senyilmaz Tiebe, PhD
Editor
EMBO Reports

Referee #1:

The authors have adequately addressed the points raised in my initial review and the paper should be published without major further delays.

I only came across one issue not identified in my original review, which can easily be fixed by a textual change. In the discussion (line 280) the authors state: "Examination of the literature reveals that most studies typically treat cells with CK-666 rather than CK-869, presumably because the latter is more toxic as it will result in complete loss of Arp2/3 function." This statement contradicts their own results showing that CK-869 cannot inhibit Arp3B containing complexes as also summarized a few sentences above. Hence, an important take home message for the cell biology community is that none of the two inhibitors can be expected to completely inhibit Arp2/3 function. Claiming otherwise at any point in the discussion distorts the central insight of the paper in my opinion.

Referee #2:

I have carefully reviewed the revised manuscript, including the author's responses to the concerns of all the original reviews. I am happy to confirm that the authors have adequately addresses my concerns and suggestions. The addition of new experiments and analyses has significantly strengthen the manuscript. I am satisfied with the revisions made, and therefore believe that manuscript is well-suited for publication in EMBO reports.

Referee #3:

The authors have corrected the manuscript as suggested and it is ready for publication.

Rev_Com_number: RC-2023-02277

New_manu_number: EMBOR-2024-59260V1-T

Corr_author: Cao

Title: CK-666 and CK-869 differentially inhibit Arp2/3 iso-complexes

All editorial and formatting issues were resolved by the authors.

Dr. Michael Way
The Francis Crick Institute
Cellular Signalling and Cytoskeletal Function
1 Midland Road
London NW1 1AT
United Kingdom

Dear Michael,

Thank you for submitting your revised manuscript. I have now looked at everything and all is fine. Therefore, I am very pleased to accept your manuscript for publication in EMBO Reports.

Congratulations on a nice work!

Kind regards,

Deniz
--
Deniz Senyilmaz Tiebe, PhD
Editor
EMBO Reports
